# T cell deficiency precipitates antibody evasion and emergence of neurovirulent polyomavirus

Matthew D Lauver[1], Ge Jin[1], Katelyn N Ayers[1], Sarah N Carey[1], Charles S Specht[2], Catherine S Abendroth[2], Aron E Lukacher[1]*

[1]Department of Microbiology and Immunology, Pennsylvania State University, Hershey, United States; [2]Department of Pathology and Laboratory Medicine, Penn State Milton S. Hershey Medical Center, Hershey, United States

**Abstract** JC polyomavirus (JCPyV) causes progressive multifocal leukoencephalopathy (PML), a life-threatening brain disease in immunocompromised patients. Inherited and acquired T cell deficiencies are associated with PML. The incidence of PML is increasing with the introduction of new immunomodulatory agents, several of which target T cells or B cells. PML patients often carry mutations in the JCPyV VP1 capsid protein, which confer resistance to neutralizing VP1 antibodies (Ab). Polyomaviruses (PyV) are tightly species-specific; the absence of tractable animal models has handicapped understanding PyV pathogenesis. Using mouse polyomavirus (MuPyV), we found that T cell deficiency during persistent infection, in the setting of monospecific VP1 Ab, was required for outgrowth of VP1 Ab-escape viral variants. CD4 T cells were primarily responsible for limiting poly-omavirus infection in the kidney, a major reservoir of persistent infection by both JCPyV and MuPyV, and checking emergence of these mutant viruses. T cells also provided a second line of defense by controlling the outgrowth of VP1 mutant viruses that evaded Ab neutralization. A virus with two capsid mutations, one conferring Ab-escape yet impaired infectivity and a second compensatory mutation, yielded a highly neurovirulent variant. These findings link T cell deficiency and evolution of Ab-escape polyomavirus VP1 variants with neuropathogenicity.

*For correspondence:
alukacher@pennstatehealth.psu.edu

Competing interest: The authors declare that no competing interests exist.

## Editor's evaluation

Progressive multifocal leukoencepalopathy (PML) is a degenerative disease of the brain that is caused by a virus in some immunocompromised patients, especially those with T-cell deficiencies. The authors use a mouse model of this virus infection to examine the components of the immune system that determine whether or not the virus will replicate and escape B-cell control. The study addresses an important question, especially given a resurgence in PML in recent years due to increasing use of immunomodulatory monoclonal antibodies to treat various diseases. The conclusions are supported by the data and confirm the expected critical role of T-cells in controlling viral early replication. The correlation of this early T-cell control with viral mutations in B-cell epitopes clarifies the relationship between proximal and distal disease causality.

## Introduction

Antibodies (Ab) are critical components of immune defense against viral pathogens and key mediators of immune control during persistent infections. Evasion of antibodies, particularly in settings of limited antibody diversity, is a potent selective pressure for viral mutations. Notably, outgrowth of escape variants is an Achilles' heel for monoclonal antibody (mAb) antiviral intervention (***Bar et al.,***

2016; *Caskey et al., 2015*; *Caskey et al., 2017*; *Mehandru et al., 2007*; *Toma et al., 2011*; *Trkola et al., 2005*). Viral evolution within a host gives rise to variants that become subject to selection for resistance to antiviral antibodies (*Inuzuka et al., 2018*; *Kinchen et al., 2018*; *Lynch et al., 2015*). For viruses that establish persistent infection, viral variants can accumulate over time and increase the likelihood for emergence of viruses carrying mutations that escape recognition by neutralizing Ab. Polyomaviruses (PyV) persist lifelong in their hosts in a 'smoldering' infectious state, a lifestyle conducive for viral evolution even by a DNA virus that commandeers the high-fidelity host cell DNA polymerase. Indeed, the human BK and JC polyomaviruses have been shown to exist as quasispecies (*Luo et al., 2012*; *Takahashi et al., 2016*; *Van Loy et al., 2015*). Whether PyV variants are sculpted by antiviral Abs to escape humoral immunity is an open question.

JCPyV causes progressive multifocal leukoencephalopathy (PML), a frequently fatal demyelinating brain disease associated with T cell immunosuppression resulting from HIV/AIDS, organ transplant immunosuppressants, and certain chemotherapeutic and immunomodulatory therapies (*Cortese et al., 2021*; *Pavlovic et al., 2018*). Typically identified by MRI following the onset of neurologic symptoms, PML is diagnosed at a point when limited treatment options exist and patients suffer permanent CNS damage. Defining the early stages of disease involved in the transition of JCPyV from a kidney to brain pathogen would facilitate earlier detection and therapeutic intervention before the development of neurologic disease. Prolonged immune suppression is a necessary antecedent for PML. For instance, high risk for natalizumab-associated PML involves infusion therapy for >24 months and a history of immunosuppression (*Berger and Fox, 2016*; *Bloomgren et al., 2012*; *Fox and Rudick, 2012*). This long period of immune suppression before clinical disease manifests raises the possibility that variants of JCPyV, including those with neurovirulent potential, may emerge over time. Iatrogenic T cell and B cell ablation therapies have independently been associated with PML. Lacking is a conceptual connection between T cell deficiency, B cell deficiency, and prolonged immune suppression in JCPyV evolution that could lead to outgrowth of neurovirulent JCPyV variants.

PML is characterized by emergence of mutations in the major capsid protein, VP1, of JCPyV, which are not found in the circulating (i.e., archetype) strains (*Gorelik et al., 2011*; *Zheng et al., 2005b*; *Zheng et al., 2005a*). The nonenveloped icosahedral viral capsid is comprised of the VP1 protein assembled into 72 pentamers. VP1 mediates attachment to cellular receptors via its four solvent-exposed loops; these loops are also the dominant targets of the host's neutralizing Ab response (*Buch et al., 2015*; *Lindner et al., 2019*; *Neu et al., 2010*). JCPyV-PML VP1 mutations, which are situated in these loops, mediate resistance to neutralizing antibodies, but also alter receptor binding and viral tropism (*Geoghegan et al., 2017*; *Gorelik et al., 2011*; *Jelcic et al., 2015*; *Lauver et al., 2020*; *Maginnis et al., 2013*; *Ray et al., 2015*). Sera from PML patients, as well as some healthy individuals, fail to neutralize particular JCPyV-PML VP1 mutations, despite these individuals having high antibody titers against WT JCPyV (*Ray et al., 2015*). The conditions that lead to the outgrowth of these variants in PML patients remain poorly understood, in part due to the lack of animal models for studying the early stages of JCPyV pathogenesis.

Because PyVs only replicate in their host species, we developed mouse polyomavirus (MuPyV) as a model for JCPyV pathogenesis and immunity (*Ayers et al., 2021*). MuPyV shares several features with JCPyV, including asymptomatic disease in immunocompetent hosts, the kidney as a dominant reservoir of virus persistence, and control by virus-specific T cells. Although both viruses bind to cell surface sialylated glycan receptors, the attachment receptor for JCPyV is the oligosaccharide lactoseries tetrasaccharide c, whereas MuPyV binds the gangliosides GD1a and GT1b (*Buch et al., 2015*; *Neu et al., 2010*; *Tsai et al., 2003*).

We previously identified a neutralizing mAb against MuPyV VP1 (*Swimm et al., 2010*). The epitope for this mAb is the dominant target of the endogenous VP1 Ab response in mice. Importantly, this mAb failed to recognize several VP1 mutations introduced into WT MuPyV that mapped to mutations seen in JCPyV isolates from PML patients (*Lauver et al., 2020*). In this study, we utilized passive immunization with this MuPyV VP1 mAb in mice lacking endogenous Abs to model the emergence of VP1 variant viruses under conditions of a limited antiviral Ab response. We found that T cell loss during persistent infection in these mice led to outgrowth of MuPyV variants carrying mutations in similar regions of the external VP1 loops to those seen in PML patients. Although an array of VP1 mutations arose in the kidneys in T-cell-depleted mice, the primary driver selecting replication-competent mutant viruses, including rare neuropathogenic variants, was evasion of Ab neutralization. With the

MuPyV infection model, we provide evidence that T cell deficiency, coupled with limited Ab coverage of VP1 epitopes, are both required for outgrowth of Ab-escape viral variants, including those with a potential for neurovirulence.

## Results

### T cell deficiency during persistent infection leads to outgrowth of antibody-resistant virus variants

We demonstrated that Ab-escape VP1 mutations emerge after serial passage of wild type (WT) MuPyV (strain A2) in host cells in the presence of a VP1-specific neutralizing mAb (*Lauver et al., 2020*). To ask whether VP1 antibody-escape mutations similarly arise in vivo, we passively immunized B-cell-deficient µMT mice with this VP1-specific mAb (clone 8A7H5) to model an inability to neutralize a frequent VP1 mutation seen among the JCPyV VP1 mutations in PML patients (*Ray et al., 2015*). µMT mice infected subcutaneously (s.c.) via the hind footpad developed high-titer chronic viremia, which was efficiently controlled by injection of VP1 mAb starting at 4 days post-infection (dpi) and continuing weekly thereafter (*Figure 1—figure supplement 1A–B*). At 20 days post-infection (dpi), sera of VP1 mAb-treated µMT mice displayed similar neutralizing titers as B6 mice against WT MuPyV, but an inability to neutralize a published VP1 mutant virus with a deletion of aspartic acid at position 295, A2.Δ295. A virus carrying this mutation was previously isolated following passage of WT virus in the presence of the VP1 mAb and identified as being resistant to neutralization (*Lauver et al., 2020*; *Figure 1—figure supplement 1C–E*). At 30 dpi, no significant differences were observed in the numbers of kidney infiltrating T cells in WT and VP1-mAb-treated µMT mice (*Figure 1—figure supplement 2A–B*).

We first investigated the effect of T cell deficiency on virus control. CD4 and/or CD8β T-cell-depleting antibodies or control rat IgG were administered to VP1 mAb-treated µMT mice beginning at 20 dpi (*Figure 1A*). Ten days after starting T cell depletion, CD4 and CD8 T cells were absent from the blood of depleted mice and virus levels were significantly elevated in the kidney, the dominant site of MuPyV persistent infection, (*Figure 1B* and *Figure 1—figure supplement 3*). In mice receiving CD4 or CD8β T-cell-depleting antibodies individually, kidney virus levels were increased in CD4, but not CD8β, T-cell-depleted mice (*Figure 1C*). In dual CD4 and CD8 T-cell-depleted mice, anti-VP1 immunofluorescence staining was localized to disrupted tubules expressing Tamm-Horsfall protein (THP), identifying the distal convoluted tubules as a major site of MuPyV replication in the kidney (*Tokonami et al., 2018*; *Figure 1D*).

Next, we asked how T cell loss affected long-term virus control. Mice were treated with combined CD4 and CD8β T-cell-depleting antibodies, CD4 T-cell-depleting antibody, CD8β T-cell-depleting antibody, or control rat IgG (*Figure 2—figure supplement 1*). Blood was collected every 20 days and screened for infectious virus by plaque assay (*Figure 2A*). Viremia became detectable in CD4 and CD8β T-cell-depleted mice as early as 40 days after dual CD4 and CD8 T cell depletion, with all mice developing viremia by approximately 100 days post depletion. CD4 T cell loss alone also led to viremia in all mice by 120 days post depletion (*Figure 2B*). In contrast, fewer than half of persistently infected mice given control IgG or CD8β T-cell-depleting antibody became viremic; in IgG-treated mice the development of viremia was delayed and peaked at approximately 100-fold lower levels than the T cell-depleted mice (*Figure 2B*). Mice receiving combined CD4 and CD8 T cell depletion or CD4 T cell depletion alone showed high systemic viral infection in the kidney, spleen, and brain (*Figure 2C–E*).

Sequencing plaque-purified virus from the blood of each viremic mouse revealed mutations in VP1, with typically only one VP1 variant found in individual mice (*Supplementary file 1*). In two mice, two mutant viruses were isolated from the mouse that each had a deletion of histidine 297 in addition to another unique mutation (E68K or N149K in one mouse and Δ147–148 or D295N in the other mouse). In a third mouse, two mutant viruses were isolated with a deletion of aspartic acid 295 and one of two mutations in the BC loop (I79S or N80K). This finding mirrors evidence that a PML patient typically harbors a single VP1 mutant JCPyV in their CSF, brain, and blood (*Gorelik et al., 2011*; *Reid et al., 2011*; *Zheng et al., 2005a*). Several of these non-synonymous single nucleotide substitutions and codon deletions were previously isolated from serially passaged MuPyV refractory to VP1 mAb-mediated neutralization, including Δ295, the most frequently detected mutation, as well as V296F, previously identified as the MuPyV equivalent of the common JCPyV PML mutation S268F (*Lauver*

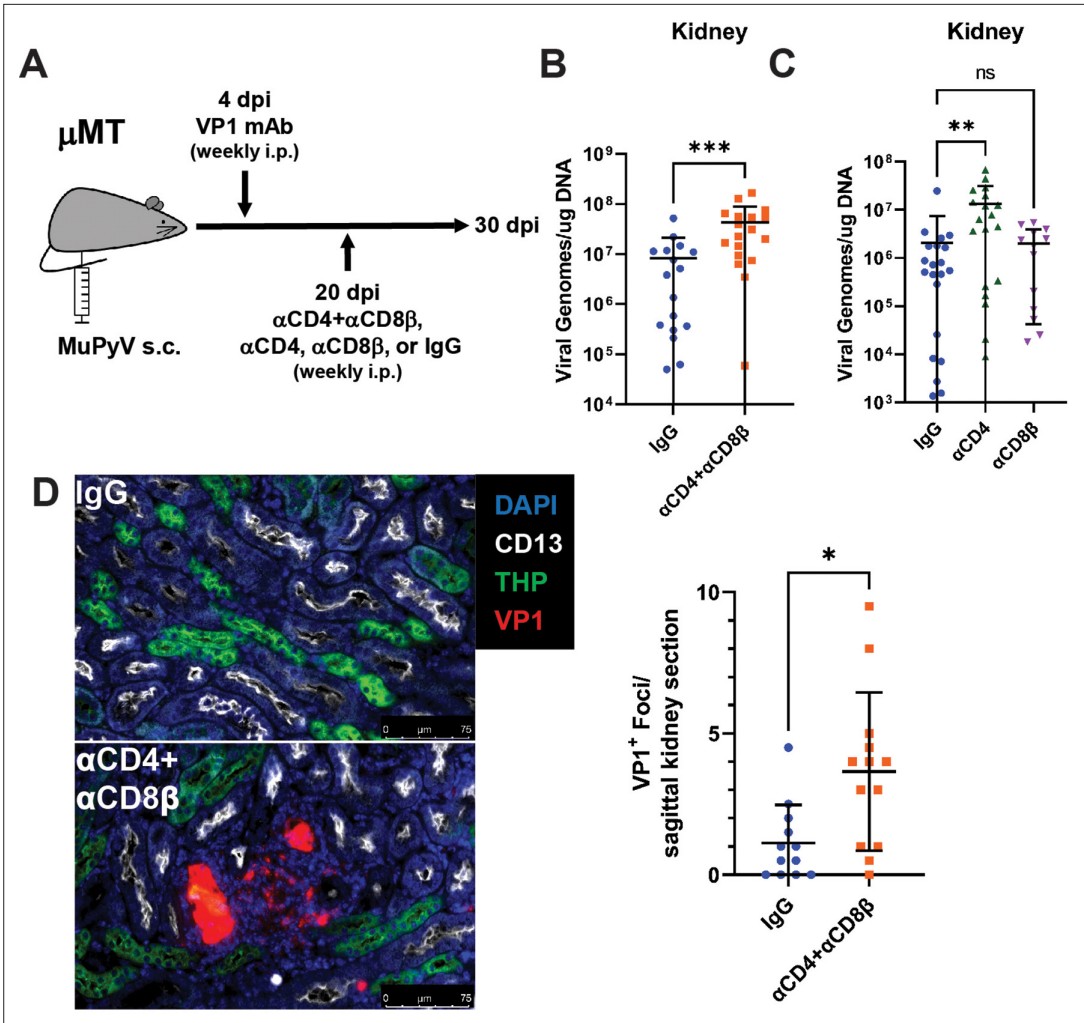

**Figure 1.** Increased MuPyV replication in the kidney following T cell depletion in VP1 mAb-treated µMT mice.
(**A**) Experimental approach for VP1 mAb treatment and T cell depletion. (**B**) Viral DNA levels in the kidney 10 days
post T cell depletion with combined αCD4 and αCD8β. Viral DNA was quantified by qPCR and compared to a
standard curve (n=16–18). (**C**) Viral DNA levels in the kidney 10 days post depletion with αCD4 or αCD8β. Viral
DNA was quantified by qPCR and compared to a standard curve (n=13–21). (**D**) (Left) Foci of virus replication
the kidney cortex 10 days post T cell depletion. Kidneys were stained for CD13 (white), THP (green) and VP1
(red). (Right) Quantification of virus foci in the kidney. Data are the average of two kidney sections per mouse
(n=12–13). Error bars are mean ± SD. Data are from at least two independent experiments. Data were analyzed
by Mann-Whitney U test (**B, D**) or Kruskal-Wallis test with Dunn's multiple comparisons test (**C**). *p<0.05, **p<0.01,
***p<0.001.

The online version of this article includes the following source data and figure supplement(s) for figure 1:

**Source data 1.** Data for the graphs in the figure.

**Figure supplement 1.** Passive immunization with a VP1 mAb in B cell-deficient mice neutralizes WT virus but not a
VP1 mutant virus.

**Figure supplement 1—source data 1.** Data for the graphs in the figure.

**Figure supplement 2.** Kidney T cell infiltrates in WT and VP1-mAb treated µMT mice.

**Figure supplement 2—source data 1.** Data for the graphs in the figure.

**Figure supplement 3.** Confirmation of T cell depletion.

**Figure supplement 3—source data 1.** Data for the graphs in the figure.

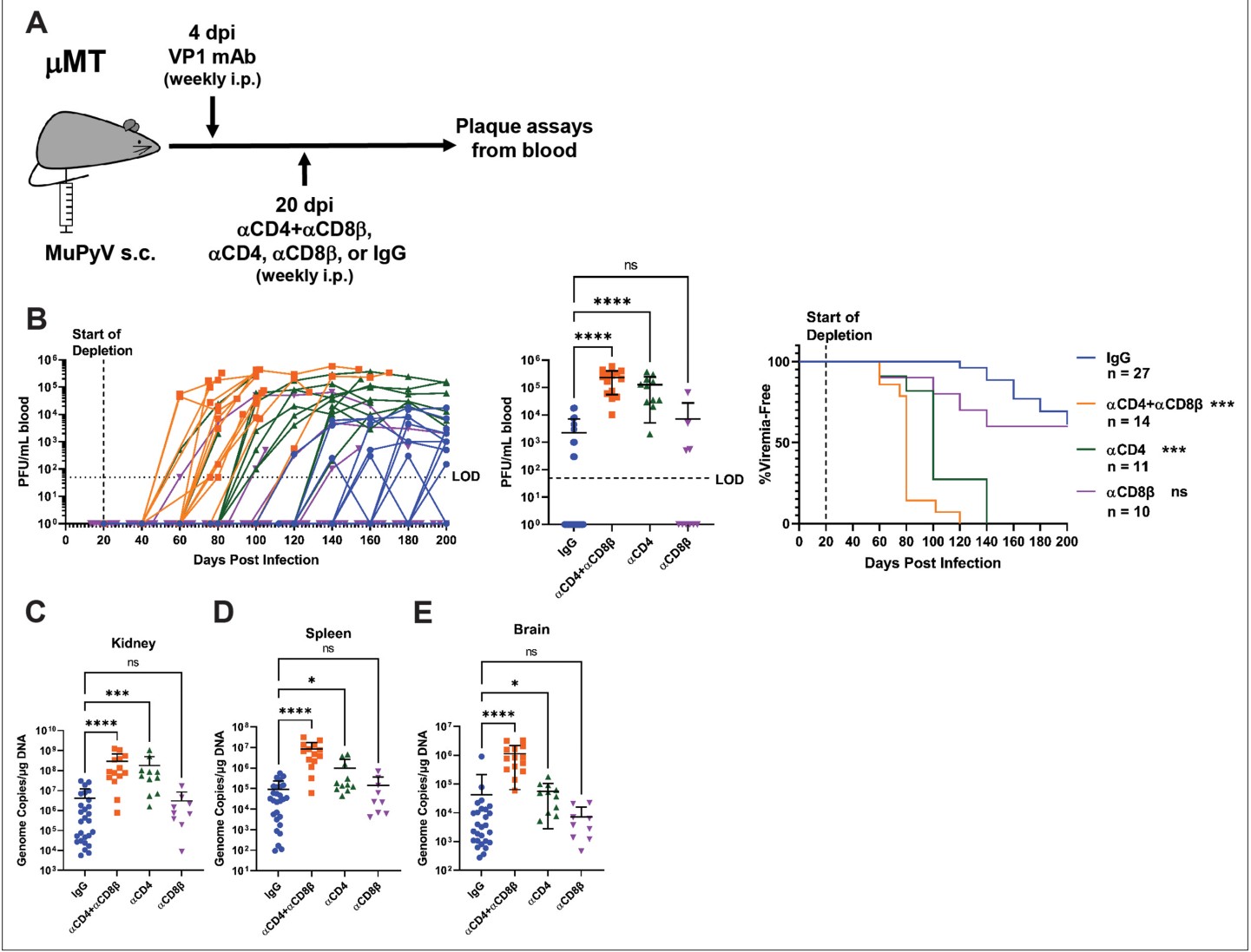

**Figure 2.** T cell loss enables viremia. (**A**) Experimental scheme for VP1 mAb treatment, T cell depletion, and detection of viremia. (**B**) (Left) Viral titers in the blood of control and T cell depleted mice over time. Viremia was measured by plaque assay from whole blood. (Center) Peak levels of viremia detected in control and T cell depleted mice. (Right) Time to development of viremia in control and T-cell-depleted mice. Indicated significances are with comparison to the IgG group. LOD: Limit of detection (n=10–27). (**C–E**) Viral DNA levels in the kidney (**C**), spleen (**D**), and brain (**E**) at the time of euthanasia in control and T-cell-depleted mice. Viral DNA was quantified by qPCR and compared to a standard curve (n=9–27). Error bars are mean ± SD. Data are from at least two independent experiments. Data were analyzed by Kruskal-Wallis test with Dunn's multiple comparisons test (B Center), (**C–E**) or Mantel-Cox test with Bonferroni's correction for multiple comparisons (B Right). *p<0.05, ***p<0.001, ****p<0.0001.

The online version of this article includes the following source data and figure supplement(s) for figure 2:

**Source data 1.** Data for the graphs in the figure.

**Figure supplement 1.** Confirmation of T cell depletion.

**Figure supplement 1—source data 1.** Data for the graphs in the figure.

et al., 2020; **Sunyaev et al., 2009**). Notably, the codon for phenylalanine in V296F in this host-derived MuPyV differs from the one we previously created by site-directed mutagenesis (**Lauver et al., 2020**). We also identified several new mutations, including combined single amino acid deletions and substitutions. Each of the VP1 mutant viruses contained a mutation in the HI loop, where the heavy chain of the VP1 mAb contacts multiple residues (**Figure 3A**; **Lauver et al., 2020**). To exclude possible effects of other mutations in the viral genome, we introduced several of these single and dual VP1 mutations into WT MuPyV using site-directed mutagenesis. We found that these mutations blocked neutralization by the VP1 mAb (**Figure 3B**). To examine the effects of the mutations on tropism, we

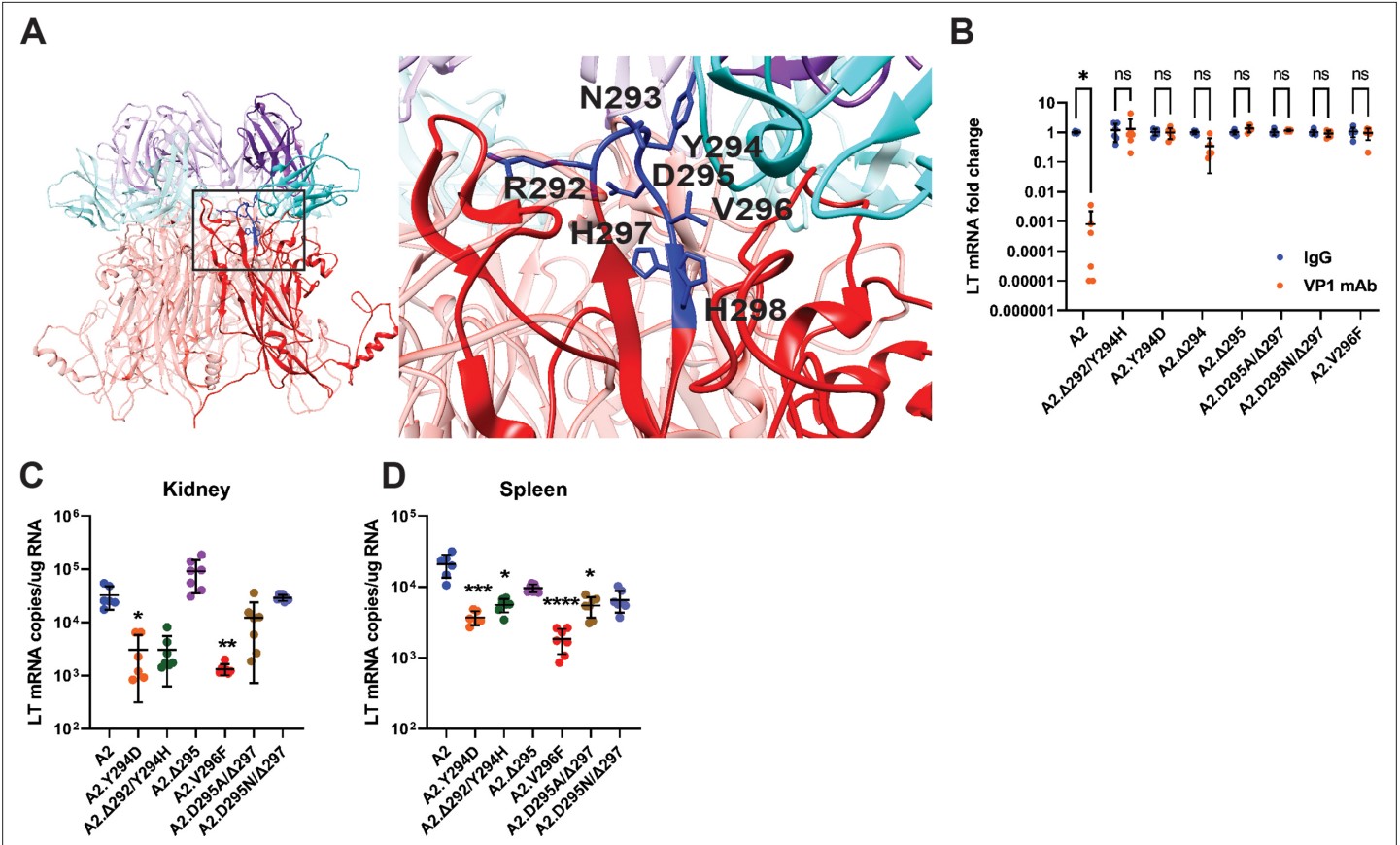

**Figure 3.** Viremia is mediated by VP1 Ab escape mutations. (**A**) Location of VP1 mutations (blue) in the HI loop of one copy of VP1 relative to the location of VP1 mAb (cyan/purple) in the Cryo-EM structure of WT VP1 and the VP1 mAb. PDB ID: 7K22 (*Lauver et al., 2020*). (**B**) VP1 mAb neutralization of VP1 mutant viruses. Viruses were preincubated with 10 μg VP1 mAb or control IgG for 30 min prior to addition to $1\times10^5$ NMuMG epithelial cells. A2 was diluted to an MOI of 0.1 PFU/cell, mutant viruses were diluted to match A2 by genomic equivalents (g.e.). Viral LT mRNA levels were quantified 24 hpi and normalized for each virus to infection with control IgG (n=6). (**C–D**) Viral mRNA levels in the kidney (**C**) and spleen (**D**) 4 dpi with VP1 mutant viruses compared to WT. WT mice were infected i.v. with $1\times10^6$ PFU of A2 or mutant viruses matched by g.e. Viral LT mRNA levels were quantified by qPCR and compared to a standard curve (n=6–7). Error bars are mean ± SD. Data are from at least two independent experiments. Data were analyzed by Mann-Whitney U test with Holm-Šídák correction for multiple comparisons (**B**) or Kruskal-Wallis test with Dunn's multiple comparisons test (**C–D**). *p<0.05, **p<0.01, ***p<0.001, ****p<0.0001.

The online version of this article includes the following source data for figure 3:

**Source data 1.** Data for the graphs in the figure.

infected mice with the mutant viruses intravenously (i.v.). The viruses were injected i.v. to examine tropism when virus is spreading in the blood, which was the condition under which these mutations were identified, and to avoid the possibility of the viruses having impaired spread from the site of s.c. inoculation. Despite the shared resistance to neutralization, the mutations had varying effects on the efficiency of viral infection in the kidney and spleen (*Figure 3C–D*). These data indicate that escape from neutralizing antibody, not shifts in tissue tropism, was the selective pressure behind the emergence of VP1 mutations in vivo.

## T cells prevent outgrowth of antibody-escape mutant virus

The absent/delayed and lower viremia in the IgG-treated mice than T cell-depleted mice led us to hypothesize that T cells could prevent/restrain the outgrowth of an antibody-escape virus if one arose during persistent infection. To do this, we treated μMT mice with VP1 mAb as before, but challenged the mice with 1000 PFU of the A2.Δ295 mutant virus 2 days after starting T cell depletion (*Figure 4A*). Mice were challenged i.v. with a lower titer inoculum of the A2.Δ295 mutant virus to mimic the development of viremia with an Ab escape mutant virus. This experimental setup allowed us to separate the

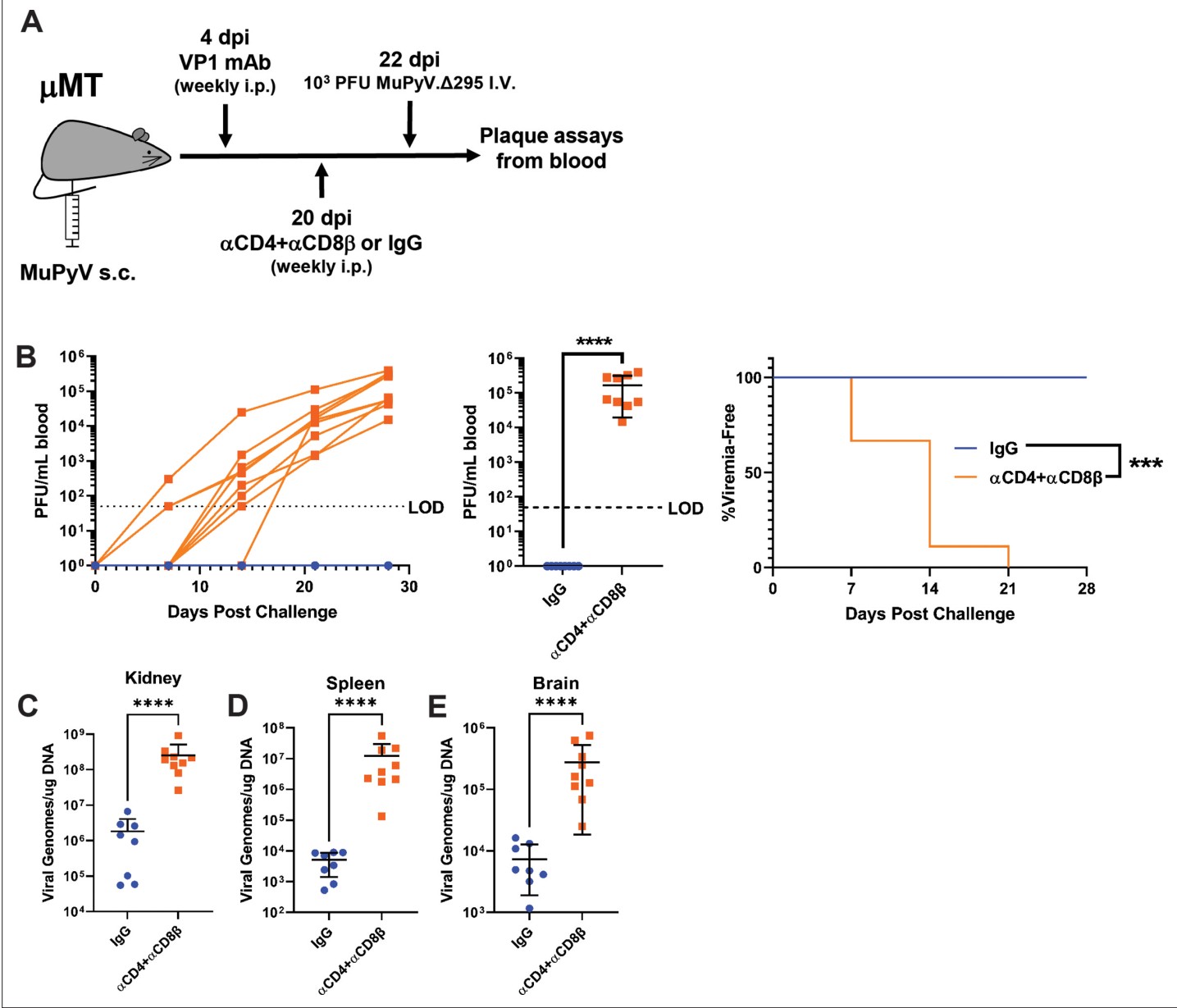

**Figure 4.** T cells prevent the outgrowth of Ab escape mutant virus. (**A**) Experimental scheme for VP1 mAb treatment, T cell depletion, A2.Δ295 challenge, and detection of viremia in μMT mice. (**B**) (Left) Viral titers in the blood of control and T-cell-depleted mice over time. Viremia was measured by plaque assay from whole blood. (Center) Peak levels of viremia detected in control and T-cell-depleted mice. (Right) Time to development of viremia in control and T-cell-depleted mice in B. LOD: Limit of detection (n=8–9). (**C–E**) Viral DNA levels in the kidney (**C**), spleen (**D**), and brain (**E**) 28 days post challenge (n=8–9). Error bars are mean ± SD. Data are from two independent experiments. Data were analyzed by Mantel-Cox test (B Right) or Mann-Whitney U test (B center), (**C–E**). ***p<0.001, ****p<0.0001.

The online version of this article includes the following source data and figure supplement(s) for figure 4:

**Source data 1.** Data for the graphs in the figure.

**Figure supplement 1.** Confirmation of T cell depletion.

**Figure supplement 1—source data 1.** Data for the graphs in the figure.

function of T cells in preventing the generation of VP1 mutations from the ability of T cells to control the outgrowth of a VP1 mutant virus. Mice receiving CD4 and CD8β T-cell-depleting antibodies developed viremia with progressively increasing infectious virus titers over time; in contrast, no viremia was detected in the control IgG-treated mice (*Figure 4B* and *Figure 4—figure supplement 1*). Moreover, the T-cell-depleted mice had 100–1000-fold higher virus levels in the kidney, spleen, and brain than

the IgG-treated mice (*Figure 4C–E*). These results provide clear evidence that T cells act to prevent viremia by MuPyV variants that escape neutralizing antibody.

## A double VP1 mutation balances viral fitness vs antibody-escape

Antibody-escape viruses carrying a substitution at D295 together with a deletion of H297 (A2.D295N/Δ297) stand out because the D295 and H297 side chains face away from the interface of VP1 and the mAb (*Figure 3A*). We generated viruses individually carrying the D295N or Δ297 mutations in the WT A2 genome (A2.D295N or A2.Δ297, respectively) to define the contributions of each mutation to recognition by the VP1 mAb and potential effects on tropism. Neutralization assays with VP1 mAb showed that A2.D295N remained sensitive, but A2.Δ297 was fully resistant, to neutralization (*Figure 5A*). Similar to parental A2, A2.D295N was readily bound by the VP1 mAb; in contrast, the mAb did not bind A2.Δ297 or A2.D295N/Δ297 (*Figure 5B*). A2.D295N also showed a similar avidity profile to parental A2 with the VP1 mAb (*Figure 5C*). Taken together, these data show that D295N did not affect neutralization, suggesting that this mutation did not contribute to antibody escape. In support, we found that A2.Δ297 failed to form plaques, indicating a defect in spread (*Figure 5—figure supplement 1A*). In contrast, A2.D295N formed plaques comparably to WT virus and A2.D295N/Δ297 produced significantly smaller sized plaques. Reduced plaque size by MuPyV has been associated with increased affinity for binding host cell receptors (*Bauer et al., 1999*). By titering infectious virus output during one round of replication by plaque assay, we observed decreased virion production by A2.D295N but significantly increased replication by A2.D295N/Δ297 (*Figure 5D*). Matching virus titers by DNA genome equivalents (g.e.), we found enhanced infection by A2.D295N/Δ297, whereas A2.Δ297 had similar infectivity to A2 and A2.D295N had reduced infectivity (*Figure 5E*). Using a low multiplicity of infection (MOI) infection to track viral spread, we found that A2.Δ297 and A2.D295N both showed a significant reduction in spread compared to A2 (*Figure 5F*). This reduction in spread was not due to reduced virus production; cells transfected with equal amounts of viral DNA showed similar levels of virus output at 72 hr by A2.D295N and A2.Δ297, but significantly more virus by A2.D295N/Δ297 (*Figure 5—figure supplement 1B*). These data indicate the Δ297 mutation impaired viral spread and prevented plaque formation. In contrast, the D295N mutation impaired both virus infection and spread but remained able to form plaques over the course of the 6-day plaque assay. In combination, however, these mutations showed a restoration and even enhancement of infectivity and spread.

We next asked whether these mutations affected receptor usage. Parental A2 and A2.D295N/Δ297 showed preferential hemagglutination (HA) activity at acidic pH (*Figure 5—figure supplement 1C*). A2.D295N and A2.Δ297 exhibited poor HA activity, with A2.Δ297 showing no HA activity across the pH spectrum. As HA activity is dependent on sialic acid binding, we next examined the effect of neuraminidase pretreatment on virus binding and infection. Binding by A2, A2.Δ297, and A2.D295N/Δ297 was dependent on sialylated host cell receptors, but binding by A2.D295N was refractory to neuraminidase treatment (*Figure 5—figure supplement 1D*). Despite these differences in binding, neuraminidase pretreatment caused a significant reduction 24 hpi in LT mRNA levels with low MOI infection and T ag+ cells with high MOI infection, indicating that infection by all viruses relied predominantly on a sialic-acid-dependent pathway (*Figure 5—figure supplement 1E–F*). The dependence on sialic acid for infection, but not for binding, by A2.D295N indicates that by itself the D295N mutation mediated binding to a non-sialylated, non-productive receptor. In contrast, the A2.Δ297 single mutant showed a drastic sensitivity to neuraminidase pretreatment, with 10,000-fold reduction in mRNA production, consistent with poor receptor binding and the lack of HA activity.

We next assessed how these mutations affected receptor binding and infection in the kidney. We incubated kidney sections from uninfected mice with virus and then stained for VP1 and kidney markers for proximal (CD13+) and distal (THP+) tubules (*Baer et al., 1997*; *Tokonami et al., 2018*). Parental A2 bound to the lumen of the distal convoluted tubules (DCT) and binding was abrogated by neuraminidase pretreatment (*Figure 5G*). Similar to A2, the A2.D295N/Δ297 showed specific, neuraminidase-sensitive binding to the DCT. A2.Δ297, however, showed no detectable binding in the kidney, whereas A2.D295N showed substantial binding to the abluminal side of the tubules but little-to no-binding to the luminal side. A2.D295N binding in the kidney was insensitive to neuraminidase treatment. We then assessed virus levels in the kidneys of mice infected i.v. with the mutant viruses. Based on LT mRNA levels in the kidney at 4 dpi, both single mutant viruses were attenuated

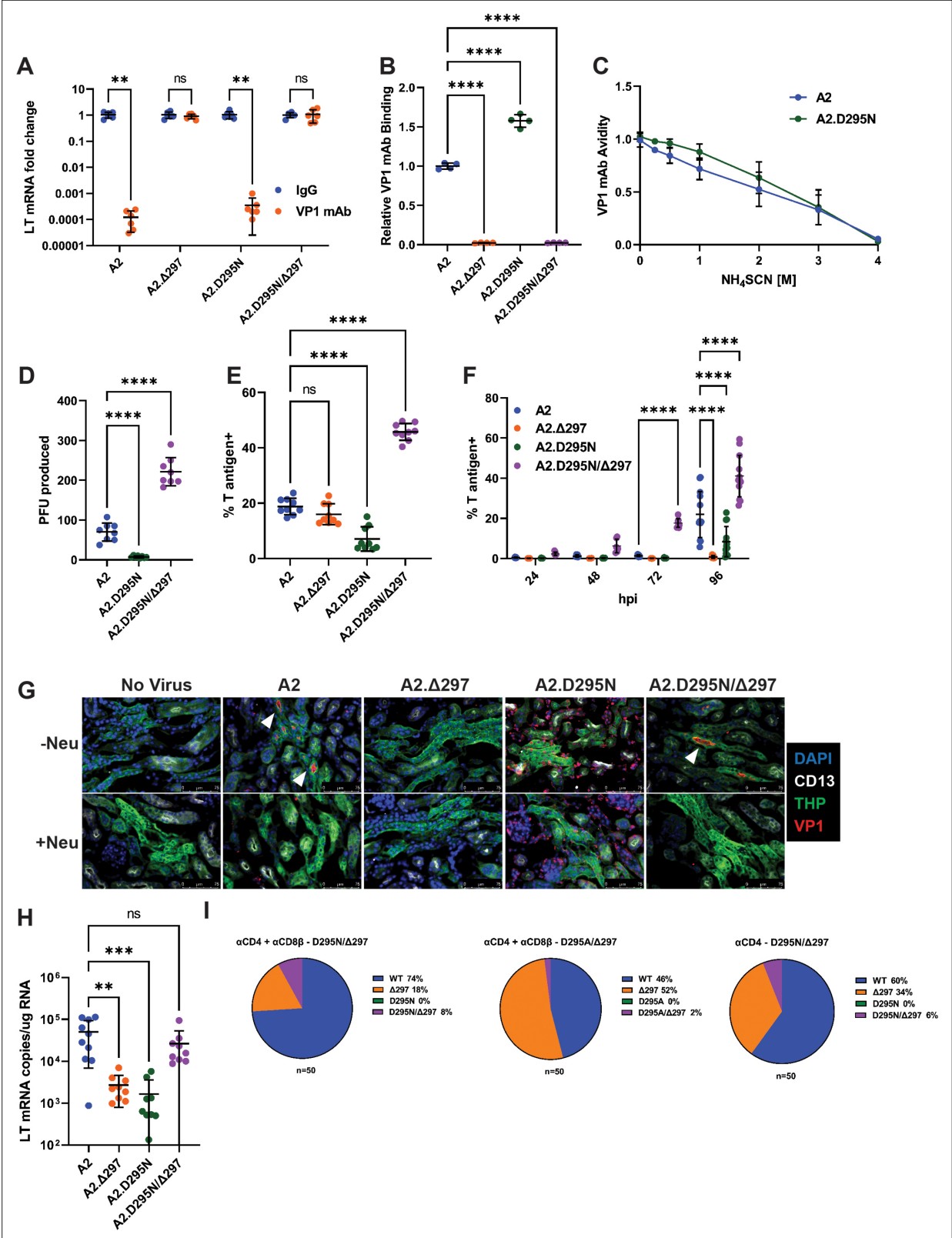

**Figure 5.** A compensatory mutation in VP1 arises to rescue defects in receptor binding caused by an Ab escape mutation. (**A**) VP1 mAb neutralization assay with D295N and Δ297 VP1 mutant viruses. Viruses were preincubated with 10 µg VP1 mAb or control IgG for 30 min prior to addition to 1x10⁵ NMuMG epithelial cells. A2 was diluted to an MOI of 0.1 PFU/cell, mutant viruses were diluted to match A2 by g.e. Viral LT mRNA levels were quantified 24 hpi and normalized for each virus to infection with control IgG (n=6). (**B**) Binding of VP1 mAb to WT or VP1 mutant viruses. Wells were

*Figure 5 continued on next page*

*Figure 5 continued*

coated with $1\times10^9$ g.e. of WT or VP1 mutant virus and incubated with VP1 mAb. VP1 mAb binding was quantified using an anti-rat secondary antibody and values were normalized to binding to WT virus (n=4). (**C**) Binding avidity of VP1 mAb for A2 and A2.D295N. VP1 mAb binding to A2 and A2.D295N was performed as in B. Prior to detection of mAb binding, virus/mAb complexes were treated with varying concentrations of $NH_4SCN$ for 15 min. Binding at each concentration was normalized to binding at 0 M $NH_4SCN$ for each virus (n=4). (**D**) Quantification of virus production in a single round of replication by plaque assay. Virus was added to $1\times10^5$ A31 cells at an MOI of 0.1 PFU/cell. Cells were lysed at 60 hpi and infectious virus was quantified by plaque assay and divided by the input virus quantity. A2.Δ297 was not included due to inability of this mutant to form plaques (n=8). (**E**) Frequency of T-antigen-positive cells 24 hpi with WT or mutant viruses. $1\times10^5$ A31 cells were infected with A2 at an MOI of 1 PFU/cell or mutant viruses matched by g.e. Cells were collected at 24 hpi, permeabilized, stained for T ag protein, and quantified by flow cytometry. (**F**) Frequency of T-antigen-positive cells at 24, 48, 72, and 96 hpi with WT or mutant viruses. $1\times10^5$ A31 cells were infected with A2 at an MOI of 0.1 PFU/cell or mutant viruses matched by g.e. Cells were collected at each time point, permeabilized, stained for T ag protein, and quantified by flow cytometry (n=6–13). (**G**) Detection of virus binding in kidney sections. PFA-fixed frozen kidney sections were treated with neuraminidase or buffer alone prior to incubation with WT or VP1 mutant virus. Sections were then stained for VP1 (red) and kidney markers [CD13 (white), THP (green)]. Neu: Neuraminidase. Representative of three independent experiments. (**H**) LT mRNA levels in the kidney 4 dpi with WT or mutant viruses. WT mice were infected i.v. with $1\times10^6$ PFU of A2 or mutant viruses matched by g.e. Viral LT mRNA levels were quantified by qPCR and compared to a standard curve (n=9–10). (**I**) Detection of D295 and H297 mutations in the kidney. VP1 sequences were PCR amplified from kidney DNA samples of the mice that developed the D295N/Δ297 and D295A/Δ297 double mutant viruses. VP1 clones were sequenced and screened for the presence of mutations at D295 and H297; the frequency of each mutation in 50 clones is shown. Error bars are mean ± SD. Data are from at least two independent experiments. Data were analyzed by Mann-Whitney U test with Holm-Šídák correction for multiple comparisons (**A**), one-way ANOVA with Dunnett's multiple comparisons test (**B and D–E**), two-way ANOVA with Dunnett's multiple comparisons test (**F**), or Kruskal-Wallis test with Dunn's multiple comparisons test (**H**). **p<0.01, ***p<0.001, ****p<0.0001.

The online version of this article includes the following source data and figure supplement(s) for figure 5:

**Source data 1.** Data for the graphs in the figure.

**Figure supplement 1.** Defects in mutant virus spread and receptor binding.

**Figure supplement 1—source data 1.** Data for the graphs in the figure.

compared to A2, whereas A2.D295N/Δ297 showed similar levels of infection to A2 (*Figure 5H*). Together, these findings indicate that the Δ297 mutation conferred resistance to VP1 mAb neutralization, but impaired viral spread and infection within the kidney. The D295N mutation alone caused aberrant receptor binding and impaired infection in vivo, but when combined with Δ297 restored proper receptor binding in the kidney during early infection.

Based on the effects of the D295N and Δ297 mutations, we hypothesized that Ab escape by the Δ297 mutation was the initial driver of mutant virus emergence in vivo, with the emergence of the mutation at D295 occurring secondary to compensate for the defect in virulence. To test this, we cloned and sequenced VP1 from the kidneys of three T cell-depleted (two CD4 and CD8β T cell depleted; one CD4 depleted) mice that developed the D295N/Δ297 and D295A/Δ297 double mutant viruses. Each of these mice rapidly became morbid after the detection of viremia necessitating their euthanasia within 30 days of the emergence of viremia. In 50 clones sequenced from the kidneys of each of the three mice, we identified sequences containing WT VP1, Δ297, and D295N/Δ297 or D295A/Δ297, but not sequences containing the D295N/A mutations alone (*Figure 5I*). The presence of the individual Δ297 mutation and absence of single D295 mutations suggests that the initial mutation was Δ297 followed by D295N/A. Although more sensitive sequencing methods may detect VP1 sequences with D295N/A mutations without Δ297, the high frequency of the Δ297 mutation strongly suggests this was the initial mutation. The rapid morbidity associated with the emergence of the mutation at D295 and subsequent viremia likely limited the accumulation of these double mutant viruses.

## Heightened neurovirulence by a VP1 double mutation virus

Given the increased replication of A2.D295N/Δ297 in vitro, we next investigated whether this virus showed altered kidney pathology in an immunocompromised host. We infected µMT mice i.v. with A2 or A2.D295N/Δ297, which resulted in chronic viremia 30 dpi by each virus (*Figure 6A*). Kidneys from A2-infected mice had foci of lymphocytic inflammation within the cortical interstitium that extended into the tubular epithelium (*Figure 6B*). In marked contrast, kidneys A2.D295N/Δ297-infected mice had only small, scattered collections of mononuclear cells. Consistent with these histopathologic differences, A2-infected kidneys also exhibited numerous, large VP1+ foci, but those in A2.D295N/Δ297-infected kidneys were far fewer and smaller (*Figure 6C–D*). To determine if A2.D295N/Δ297 was neurovirulent, we infected WT mice intracranially (i.c.), a route of inoculation which mediates efficient MuPyV infection of the brain (*Lauver et al., 2020*; *Mockus et al., 2020*). Four dpi,

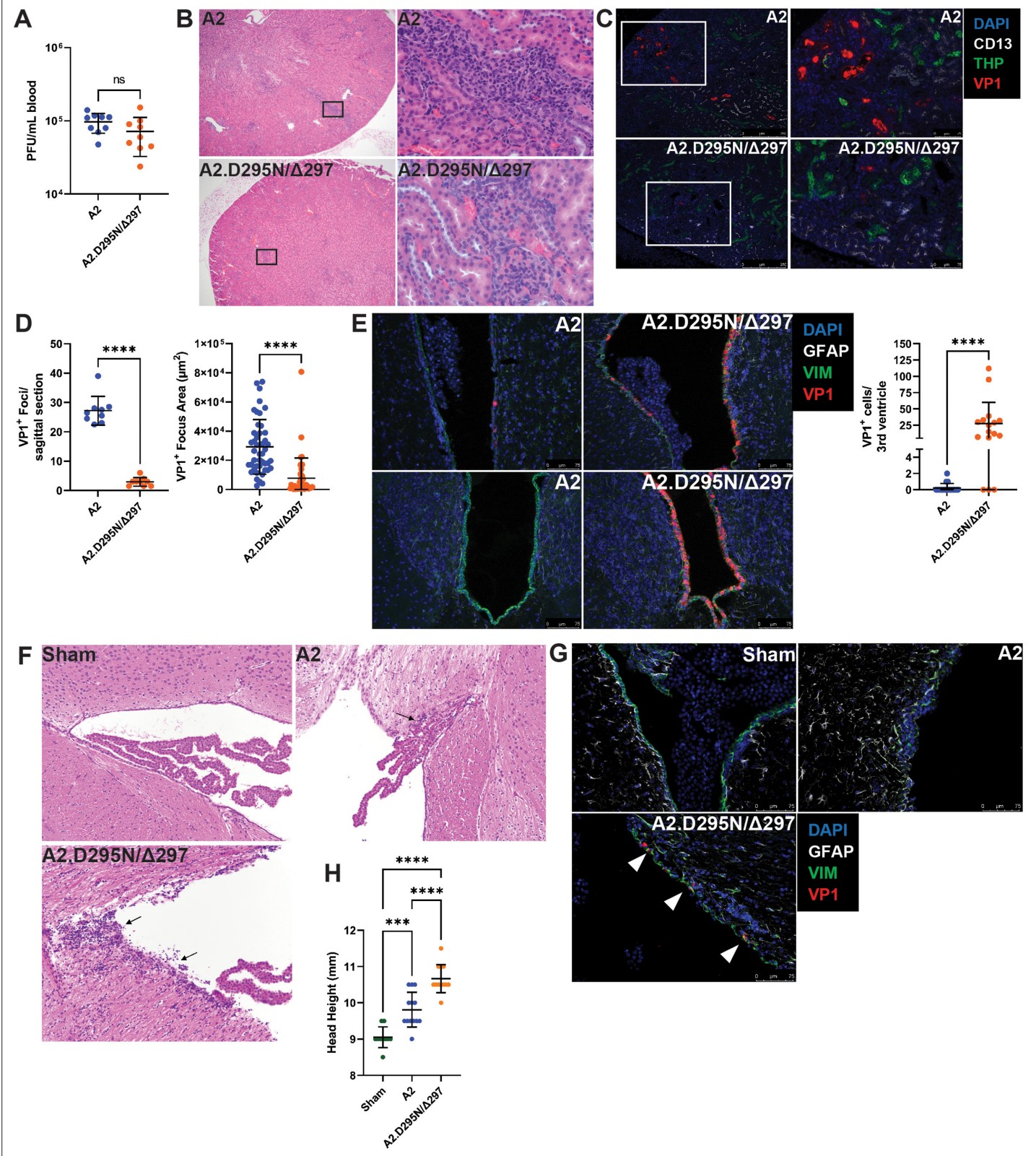

**Figure 6.** Reduced kidney tropism and heightened neurovirulence by a VP1 double mutant virus. (**A**) Viremia in μMT mice 30 dpi with A2 or A2.D295N/Δ297 i.v. Viremia was quantified by plaque assay from whole blood (n=9). (**B**) H&E stained sagittal sections of kidneys from μMT mice 30 dpi after i.v. inoculation with A2 or A2.D295N/Δ297. Left: ×50 magnification. Right: ×500 magnification. (**C**) Foci of VP1+ cells in kidneys from μMT mice 30 dpi with A2 or A2.D295N/Δ297 i.v. Kidneys were stained for CD13 (white), THP (green) and VP1 (red). (**D**) Quantification of the number (Left, n=9) and size (Right,

*Figure 6 continued on next page*

*Figure 6 continued*

n=42–45) of VP1⁺ foci in kidneys from μMT mice 30 dpi with A2 or A2.D295N/Δ297 i.v. Foci number is the average of two sagittal kidney sections per mouse. For quantifying foci area, five random foci per mouse or the maximum number of foci found were imaged and the area of each VP1⁺ focus was calculated using ImageJ. (**E**) VP1⁺ cells in the lateral (top) and third (bottom) ventricles of WT mice 4 dpi with A2 or A2.D295N/Δ297 inoculated i.c. and quantification of VP1⁺ cells in the third ventricle (n=16–18). (**F**) H&E stained coronal sections from brains of sham (top left), A2 (top right), or A2.D295N/Δ297 (bottom left) i.c.-inoculated mice 30 dpi (×200 magnification). Arrows indicate sites of ependymal inflammation. (**G**) VP1 staining in the ventricles of WT mice 30 dpi with A2 or A2.D295N/Δ297 i.c. VP1⁺ cells are indicated with white markers. (**H**) Quantification of hydrocephalus 30 dpi after i.c. inoculation with vehicle, A2, or A2.D295N/Δ297. Coronal head height was measured with a Vernier caliper in line with the ear canal to the nearest 0.5 mm (n=10–13). Error bars are mean ± SD. Data are from at least two independent experiments. Data were analyzed by Mann-Whitney U test (A, D, and E) or one-way ANOVA with Dunnett's multiple comparisons test (**H**). ***p<0.001, ****p<0.0001.

The online version of this article includes the following source data for figure 6:

**Source data 1.** Data for the graphs in the figure.

A2-infected brains showed only sparse VP1⁺ ependymal cells lining the ventricles. In contrast, brains of A2.D295N/Δ297-infected mice had extensive VP1⁺ ependymal cells (*Figure 6E*). Consistent with this dramatic difference in extent of ependymal infection, A2-infected brains at 30 dpi had only small ependymal lymphocytic aggregates, whereas A2.D295N/Δ297-infected brains contained more extensive ependymitis as well as periventricular edema (*Figure 6F*). Additionally, of the sections examined only brains from mice persistently infected with A2.D295N/Δ297 had VP1⁺ cells in the periventricular region (*Figure 6G*). Hydrocephalus is consistently seen in mice i.c. inoculated with A2 MuPyV (*Lauver et al., 2020*; *Mockus et al., 2020*). A2.D295N/Δ297-infected mice developed hydrocephalus to a significantly higher degree than A2-infected mice (*Figure 6H*). These data show that the endogenously derived A2.D295N/Δ297 VP1 mutant virus had lower tropism for the kidney, but dramatically higher capacity to infect the cerebral ventricular system than the parental A2 virus. Similarly, impaired kidney pathogenesis but retained neurovirulence is a hallmark feature of A2 virus carrying the VP1 V296F mutation, sequence-equivalent to the frequent S268F VP1 mutation in JCPyV-PML (*Lauver et al., 2020*). Thus, VP1 neutralizing Ab in T-cell-deficient hosts can select rare Ab-escape virus variants possessing high neuropathogenicity.

## Discussion

Although depressed anti-JCPyV T cell immunity is the dominant risk factor for PML, JCPyVs in the CNS of PML patients have VP1 mutations that allow evasion of antiviral antibodies (*Cortese et al., 2021*; *Jelcic et al., 2015*; *Lauver et al., 2020*; *Pavlovic et al., 2018*; *Ray et al., 2015*). To connect T cell deficiency and Ab-escape JCPyV variants, we investigated T cell insufficiency in mice infected with MuPyV in the setting of a restricted VP1 antibody response. Our findings allowed us to develop a model integrating T cell and humoral deficiencies with emergence of neurovirulent PyV, as illustrated in *Figure 7*. First, impaired T cell control results in increase PyV replication in the kidney, the central reservoir for persistent PyV infection. Elevated viral replication is accompanied by low-level, stochastic mutagenesis of the viral genome, where particular mutations in VP1 abrogate binding by VP1-specific Ab. By extension, the host must have an inherited or acquired VP1 Ab response that targets few VP1 epitopes. A subset of viruses carrying Ab-escape VP1 mutations acquire the capacity to replicate in the CNS. T cell deficiency further handicaps a second line of defense by antiviral T cells to control Ab-escape VP1 variant viruses. In summary, T cell deficiency acts at two levels to set the stage for emergence of antibody-escape VP1 mutations endowed with neurotropic potential.

The kidney is the major site of JCPyV persistence (*Berger et al., 2017*). We observed preferential binding of MuPyV to the DCT, mirroring what has been reported for JCPyV, and virus replication in the DCT following T cell loss (*Figures 1D and 5G*; *Haley et al., 2015*). This localization of virus binding to the DCT, despite differences in the sialylated glycans bound by each virus, suggests a conserved site of persistence within the kidney for these viruses. Virus binding and replication within the DCT may provide a level of local protection from neutralizing antibodies, necessitating T cell control to limit virus replication within the local kidney environment. Conditions of immune suppression deprive the kidney of this cellular immune control over PyV infection, leading to elevated virus replication there. Consistent with this are reports of depressed JCPyV-specific T cell responses in individuals treated with natalizumab and increased JCPyV shedding in the urine of immunosuppressed individuals

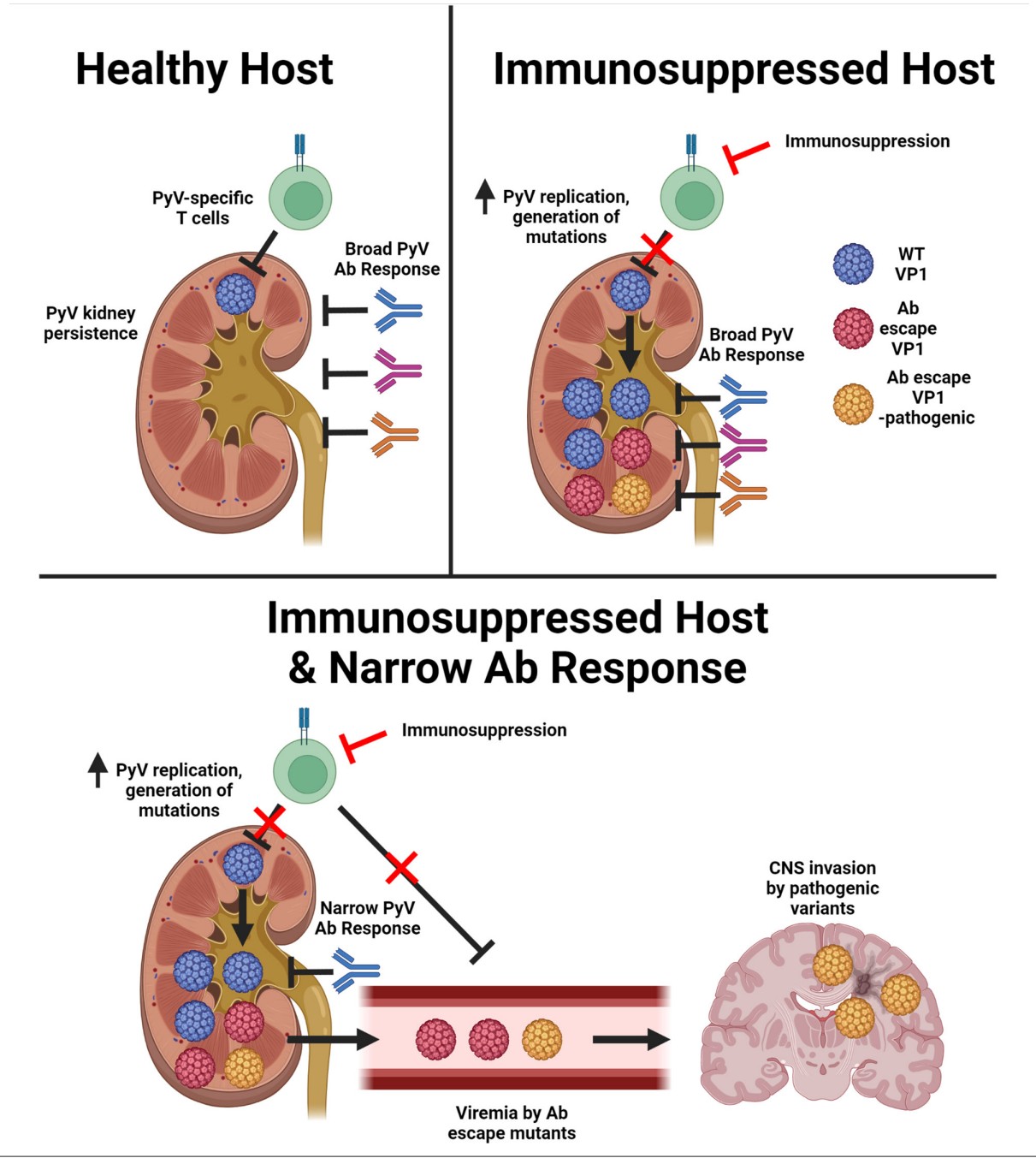

**Figure 7.** T cell deficiency and a narrow antiviral Ab response enable emergence of Ab-escape, neurovirulent PyV variants. Model for the evolution of Ab escape, neurovirulent PyV VP1 variants in the setting of T cell deficiency and a neutralizing Ab response with limited coverage of VP1 epitopes on the virus capsid. Figure was created with Biorender.com.

(*Behzad-Behbahani et al., 2004*; *Chen et al., 2009*; *Delbue et al., 2015*; *Perkins et al., 2012*). T cells are the predominant cell type responsible for the production of interferon gamma (IFN-γ), which limits BKPyV replication in kidney cells and is necessary for effective control of MuPyV kidney infection (*Abend et al., 2007*; *Byers et al., 2007*; *Wilson et al., 2011*). T cell loss during PyV infection thus eliminates the major source of IFN-γ and virus control.

PML is associated with conditions of CD4 T cell impairment, including HIV/AIDS and idiopathic CD4 T cell lymphopenia (*Berger et al., 1987*; *Berger et al., 1998*; *Pavlovic et al., 2018*). Decreases in CD4, but not CD8, T cells are seen in the CSF of natalizumab-treated MS patients and CD4 T cell

epitope-escape JCPyVs have been found in PML patients (*Jelcic et al., 2016*; *Schneider-Hohendorf et al., 2014*; *Stüve et al., 2006*). We observed higher kidney virus levels at 10 days post depletion and a higher frequency of viremia in CD4 T cell-depleted mice than in CD8 T cell-depleted mice (*Figures 1C and 2B*). CD4 T cells can exert direct antiviral effector activities as well as provide help to maintain optimal virus-specific CD8 T cell responses in nonlymphoid organs, including the brain and lung (*Ren et al., 2020*; *Son et al., 2021*). Tissue-resident CD4 T cells have also been shown to promote development of protective B cell responses in the lungs of influenza virus-infected mice (*Son et al., 2021*). It will be important to elucidate how CD4 T cells restrict PyV replication in the kidneys and viremia by Ab-escape variant viruses.

Utilizing the host DNA replication machinery, PyVs are regarded as having highly stable genomes with a low incidence of mutations (*Sanjuán and Domingo-Calap, 2016*). Clinical and experimental data, however, indicate that mutations arise during the course of PyV infection. JCPyV and BKPyV isolates from PML and polyomavirus-associated nephropathy patients, respectively, are characterized by mutations in both the VP1 capsid protein and the non-coding control region (NCCR) (*Boldorini et al., 2009*; *Gorelik et al., 2011*; *Krautkrämer et al., 2009*; *Peretti et al., 2018*; *Reid et al., 2011*; *Tremolada et al., 2010*; *Vaz et al., 2000*; *Zheng et al., 2005b*; *Zheng et al., 2005a*). Likewise, rearrangements in the NCCR of BKPyV emerge after extended passage in vitro, and VP1 mutations arise in BKPyV and MuPyV after serial passage in the presence of neutralizing mAb (*Lauver et al., 2020*; *Lindner et al., 2019*; *Zhao and Imperiale, 2021*). These data suggest that conditions favoring certain mutations (e.g., mutations enhancing replication or enabling antibody evasion) can promote the emergence and outgrowth of viruses with particular genomic alterations.

Several mechanisms provide avenues for VP1 mutations. Missense mutations may represent failed attempts by the host cell to restrict viral replication. The APOBEC3 family of proteins are cytosine deaminases that induce mutagenic damage in the genomes of DNA viruses as a mechanism of anti-viral defense. APOBEC3B is upregulated during BKPyV nephritis and has been implicated in the emergence of BKPyV VP1 mutations in kidney transplant recipients and mutations in the Merkel cell PyV genome (*Peretti et al., 2018*; *Que et al., 2021*). The presence of deletion mutations could represent the remnants of double-stranded breaks repaired by nonhomologous end joining (NHEJ). DNA double strand breaks occur during PyV infection, and aspects of the DNA damage response pathway are required for efficient PyV replication (*Erickson and Garcea, 2019*; *Heiser et al., 2016*; *Jiang et al., 2012*; *Justice et al., 2019*; *Sowd et al., 2013*). The NCCR of BKPyV undergoes significant recombination events resulting from NHEJ during replication in vitro (*Zhao and Imperiale, 2021*). It is possible that NHEJ repair of breaks in VP1 during replication results in a variety of recombinations and small deletions. Those that are tolerated by the virus and confer a selective advantage, in this case in-frame deletions conferring Ab escape, are thus able to emerge and be detected.

JCPyV VP1 mutations identified in PML patients are associated with impaired receptor binding and infectivity. Only archetype JCPyV is found in the urine of PML patients, despite the presence of VP1 mutant virus in the blood and CSF of these patients (*Geoghegan et al., 2017*; *Gorelik et al., 2011*; *Maginnis et al., 2013*; *Reid et al., 2011*; *Zheng et al., 2005a*). Here, we find that the selective pressure of antibody escape drove outgrowth of VP1 mutants with varying kidney infectivity, implicating immune evasion, rather than tropism, as the primary driver of VP1 mutations under these conditions (*Figure 3B–D*). The initial emergence of the Δ297 mutation, despite imparting severe defects in kidney tropism, indicates that Ab escape can promote the outgrowth of otherwise crippled PyV variants, a feature seen in several other chronic viral infections (*Kalinina et al., 2003*; *Kinchen et al., 2018*; *Lynch et al., 2015*).

The defect in spread and infectivity in vivo by the Δ297 mutation readily selected for a variety of secondary point mutations that restored or even enhanced virulence, generating viruses able to efficiently spread in vitro. The point mutations found together with the Δ297 mutation all involved a positive net change in charge (*Supplementary file 1*). These mutations are all located near the receptor binding pocket, and most of the mutated side chains face toward the location of receptor binding. With the exception of E91, however, these residues have not been reported to be involved in receptor binding (*Buch et al., 2015*; *Stehle and Harrison, 1997*). Mutation of E91 to glycine causes impaired infection by promoting binding to decoy pseudoreceptors (*Bauer et al., 1999*; *Buch et al., 2015*; *Qian and Tsai, 2010*). Similarly, the D295N mutation alone caused pseudoreceptor binding that impaired infectivity in vitro and in vivo (*Figure 5D–H*). In the context of the Δ297 mutation,

however, these normally deleterious mutations synergized to restore proper receptor binding and even enhance virulence. The D295N/Δ297 double mutant displayed increased infectivity and virus production in vitro (*Figure 5D–F*). In vivo, the double mutant virus showed similar acute kidney infection to WT virus but decreased kidney infection/pathology during chronic infection (*Figures 5H and 6A–D*). This impairment of kidney pathology during chronic infection suggests a defect in persistence within the kidney by the D295N/Δ297 mutant virus. Within the brain, D295N/Δ297 caused increased acute infection of the ependyma and heightened chronic infection in the periventricular region, as well as increased CNS pathology (*Figure 6E–G*). Retention of neurotropism and a loss of kidney tropism was previously seen with the V296F MuPyV VP1 mutant and may be a common feature of PML-associated JCPyV mutations (*Lauver et al., 2020*). This increased CNS virulence could stem from more efficient receptor binding and release, leading to increased viral dissemination throughout the ventricular system and elevated infection of the ependyma during persistence. Increased spread and pathogenesis resulting from altered receptor binding has been seen with a separate MuPyV HI loop mutation, V296A (*Bauer et al., 1995*; *Bauer et al., 1999*; *Buch et al., 2015*). As both the D295 and H297 residues face into the pore at the center of each VP1 pentamer, these mutations may facilitate more efficient interactions with the VP2/3 minor capsid proteins, whose C-termini extend into the pore (*Chen et al., 1998*).

The mutant viruses we identified all carried at least one mutation in the HI loop, which is also the most common site of JCPyV VP1 mutations in PML patients (*Gorelik et al., 2011*; *Reid et al., 2011*). The epitope of the VP1 mAb has significant contribution from the HI loop and competes with a large portion of the endogenous antibody response generated by MuPyV-infected mice (*Lauver et al., 2020*). The dominant targets of antibodies in JCPyV-infected individuals have not been determined, but given the frequency of JCPyV mutations seen in the HI loop this region may be a common target of neutralizing Ab across species. PML patients typically carry a mutant JCPyV with a single VP1 mutation, rather than the double mutants we saw emerge in several mice (*Supplementary file 1*; *Gorelik et al., 2011*; *Reid et al., 2011*). This difference may stem from the nature of Ab escape mutation, with substitution to a bulkier side chain in JCPyV (e.g., L55F, S266F/Y, and S268F/Y) rather than the deletion of H297 seen in MuPyV. Although both types of mutations alter/impair infectivity, bulky substitutions in JCPyV may do so by occluding a portion of the receptor binding pocket, where there may be reduced opportunity for compensation by a secondary mutation. Already the smallest external loop in VP1, the HI loop of JCPyV (M261 to S268: 8 residues) is one residue shorter than the HI loop of MuPyV (W288 to V296: 9 residues) (*Liddington et al., 1991*). Thus, the HI loop of JCPyV may be too short to tolerate even a single amino acid deletion and still allow proper positioning of the H and I β strands.

Insufficient humoral control of PyV mutants opens numerous pathways for viral invasion of the CNS. The lack of neutralization could promote infection of hematopoietic cells that then traffick to the brain (*Dubois et al., 1997*; *Wollebo et al., 2015*). Alternatively, the absence of neutralizing antibody could enable virus within the vasculature of the brain to infect the cells of the blood-brain barrier (BBB) or blood-CSF barrier (BCSFB). Brain microvascular endothelial cells are susceptible to JCPyV infection, and could provide viral access into the CNS parenchyma (*Chapagain et al., 2007*). The BCSFB, formed by the choroid plexus in the brain ventricles, is another possible route of entry by the virus into the brain. JCPyV infects choroid plexus epithelial cells (*Corbridge et al., 2019*; *Haley et al., 2015*; *O'Hara et al., 2018*; *O'Hara et al., 2020*). Infection of choroid plexus epithelium would enable virus spread into the CSF, dissemination throughout the ventricular system, and access to the brain parenchyma via infection of the ependymal cells lining the ventricles. In line with this possibility is the permissivity of ependymal cells for productive MuPyV infection (*Lauver et al., 2020*; *Mockus et al., 2020*).

Our findings demonstrate a '2-hit' requirement for development of VP1 mutant PyVs: (1) a narrow VP1 antibody response; and (2) T cell insufficiency, which acts both to limit virus replication in the kidney and eliminate VP1 variants that evade antiviral antibodies. T cell loss leads to elevated virus replication that in the context of a narrow VP1 antibody response results in viremia by Ab-escape mutant viruses. Among these Ab-escape variants will be those carrying the potential for neurovirulence. The stochastic evolution of VP1 mutations, together with depressed anti-PyV T and B cell immunity, may account for the long timeframe between PML and iatrogenic immunosuppression, the array of PML-associated agents with unrelated mechanisms of action, and the rarity of PML in susceptible hosts.

# Materials and methods

**Key resources table**

| Reagent type (species) or resource | Designation | Source or reference | Identifiers | Additional information |
|---|---|---|---|---|
| Antibody | Anti-VP1 (Rat Clone 8A7H5) | *Swimm et al., 2010* | Clone 8A7H5 | 250 µg/week |
| Antibody | ChromPure Rat IgG | Jackson ImmunoResearch | Cat#012-000-003 RRID: AB_2337136 | 250 µg/week |
| Antibody | Anti-CD8β (H35-17.2) | *Golstein et al., 1982* | N/A | 250 µg/week |
| Antibody | Anti-CD4 (GK1.5) | *Dialynas et al., 1983* | N/A | 250 µg/week |
| Antibody | Anti-VP1 (Rabbit polyclonal) | Provided by Robert Garcea | N/A | IF (1:1,000) FC (1:10,000) |
| Antibody | Anti-Vimentin (Rat Clone 280618) | R&D Systems | Cat#MAB2105 RRID: AB_2241653 | IF (1:500) |
| Antibody | Anti-GFAP (Goat polyclonal) | Abcam | Cat#ab53554 RRID: AB_880202 | IF (1:1,000) |
| Antibody | Anti-THP (Rat monoclonal) | R&D Systems | Cat#MAB5175 RRID: AB_2890000 | IF (1:1,000) |
| Antibody | Anti-CD13 (Goat polyclonal) | R&D Systems | Cat#AF2335 RRID: AB_2227288 | IF (1:500) |
| Antibody | Anti-Goat IgG AF488 (Bovine polyclonal) | Jackson ImmunoResearch | Cat#805-545-180 RRID: AB_2340883 | IF (1:500) |
| Antibody | Anti-Rat IgG AF555 (Donkey polyclonal) | Abcam | Cat#ab150154 RRID: AB_2813834 | IF (1:500) |
| Antibody | Anti-Rabbit IgG AF647 (Donkey polyclonal) | Jackson ImmunoResearch | Cat#711-605-152 RRID: AB_2492288 | IF (1:500) |
| Antibody | Anti-CD8α-AF700 (Clone 53–6.7) | Biolegend | Cat#100730 RRID: AB_493703 | FC (1:200) |
| Antibody | Anti-CD45-PerCP/Cy5.5 (Clone 30-F11) | Biolegend | Cat#103132 RRID: AB_893340 | FC (1:200) |
| Antibody | Anti-CD45-FITC (Clone 30-F11) | BD | Cat#553080 RRID: AB_394610 | FC (3 µg/mouse) |
| Antibody | Anti-CD4-PE (Clone RM4-5) | Biolegend | Cat#100512 RRID: AB_312715 | FC (1:200) |
| Antibody | Anti-CD4-BV421 (clone GK1.5) | BD | Cat#562891 RRID: AB_2737870 | FC (1:200) |
| Antibody | Anti-Rat IgG-APC (Goat polyclonal) | BD | Cat#551019 RRID: AB_398484 | FC (1:200) |
| Antibody | Anti-Mouse IgG-HRP (Goat polyclonal) | Bethyl Laboratories INC | Cat#A90-116P RRID: AB_67183 | ELISA (1:7,000) |
| Other | Fixable Viability Dye eFluor 780 | ThermoFisher | Ref#65-0865-14 | FC (1:1,000) |
| Other | Flow Cytometry Absolute Count Standard | Bangs Laboratories | Cat#580 | |
| Other | MuPyV (Strain A2) | N/A | N/A | |
| Other | Sheep Red Blood Cells | Innovative Research | Cat#ISHRBC10P15ML | |
| Peptide, recombinant protein | VP1 pentamers | Provided by Robert Garcea | N/A | |
| Peptide, recombinant protein | Benzonase Nuclease | Sigma | Cat#E1014 | |
| Peptide, recombinant protein | Neuraminidase from *Vibrio cholerae* | Sigma | Cat#N6514 | IF (1:100) FC (1:200) |

*Continued on next page*

*Continued*

| Reagent type (species) or resource | Designation | Source or reference | Identifiers | Additional information |
|---|---|---|---|---|
| Peptide, recombinant protein | Collagenase (Type I) | Worthington | Cat#LS004197 | |
| Chemical compound, drug | TRIzol Reagant | ThermoFisher | Ref#15596018 | |
| Chemical compound, drug | RevertAid H Minus Reverse Transcriptase | ThermoFisher | Cat#EP0451 | |
| Chemical compound, drug | Lipofectamine 2000 Transfection Reagent | ThermoFisher | Cat#11668030 | |
| Chemical compound, drug | Lipofectamine 3000 Transfection Reagent | ThermoFisher | Cat# L3000008 | |
| Commercial assay, kit | PFHM-II Protein-Free Hybridoma Medium | ThermoFisher | Ref#12040–077 | |
| Commercial assay, kit | TOPO TA Cloning Kit | ThermoFisher | Ref#45–0641 | |
| Commercial assay, kit | TBP PrimeTime XL qPCR Assay | IDT | Mm.PT.39a.22214839 | |
| Commercial assay, kit | 1-Step Ultra TMB-ELISA | ThermoFisher | Ref#34028 | |
| Commercial assay, kit | 96 Well EIA/RIA Polystyrene High Bind Microplate | Fisher Scientific | Cat#3590 | |
| Commercial assay, kit | PerfectCTa FastMix II ROX | Quantabio | P/N 84210 | |
| Commercial assay, kit | PureLink Viral RNA/DNA mini Kit | ThermoFisher | Ref#12280–050 | |
| Commercial assay, kit | Wizard Genomic DNA Purification Kit | Promega | Ref#A1120 | |
| Commercial assay, kit | QuikChange II Site-Directed Mutagenesis Kit | Agilent | Cat#200523 | |
| Commercial assay, kit | CELLine Disposable Bioreactor | Fisher Scientific | Cat#353137 | |
| Cell Line (*M. musculus*) | BALB/3T3 Clone A31 | ATCC | CCL-163; RRID:CVCL_0184 | |
| Cell Line (*M. musculus*) | NMuMG | ATCC | CRL-1636; RRID:CVCL_0075 | |
| Strain, strain background (*M. musculus*) | C57BL/6 Mice | National Cancer Institute | Cat#OIC55 | |
| Genetic reagent (*M. musculus*) | μMT Mice | Jackson Laboratory | Cat#002288; RRID:IMSR_JAX:002288 | |
| Software, algorithm | Prism | Graphpad | v 9.3.1; RRID:SCR_002798 | |
| Software, algorithm | FlowJo | BD | v 10.6.1; RRID:SCR_008520 | |
| Software, algorithm | ImageJ | NIH | v 1.8.0; RRID:SCR_003070 | |
| Software, algorithm | Leica LAS X | Leica | v 3.7.2; RRID:SCR_013673 | |

## Materials availability

Reagents generated in this study are available from the corresponding author with a Materials Transfer Agreement.

## Mice

C57BL/6 mice were purchased from the National Cancer Institute and μMT mice were purchased from Jackson Laboratories. Mice were housed and bred under specific pathogen-free conditions. Male and female mice 6–12 weeks of age were used for experiments. All mouse experiments were approved by the Penn State College of Medicine Institutional Animal Care and Use committee.

## Cell lines

NMuMG and BALB/3T3 clone A31 ("A31") cell lines were purchased from ATCC. Cell lines were authenticated by STR profiling (ATCC), mycoplasma negative, used at low passage, and examined for

correct morphology. The 8A7H5, H35-17.2, and GK1.5 hybridomas were grown in PFHM-II Protein-Free Hybridoma Medium (Thermofisher) (*Dialynas et al., 1983*; *Golstein et al., 1982*; *Swimm et al., 2010*). mAb was produced in CELLine bioreactor flasks (Corning). All other cells were kept in Dulbecco's Minimal Eagle Media supplemented with 10% fetal bovine serum, 100 U/mL penicillin, and 100 U/mL streptomycin.

## Viruses

All experiments were done using the A2 strain of MuPyV. Viral stocks were generated by transfection of viral DNA into NMuMG cells. Mutant viruses were generated by site-directed mutagenesis of the parental A2 viral genome with forward and reverse primers for each mutation (*Supplementary file 2*). DNA was isolated from the virus stocks and sequenced to confirm the presence of the mutation. The A2.Δ294, A2.Δ295, and A2.V296F viruses were generated previously (*Lauver et al., 2020*).

## Virus titering and sequencing

Viruses were titered by plaque assay on A31 fibroblasts or by qPCR for encapsidated genomes (*Lukacher and Wilson, 1998*). For genome titering, 1 µL of virus lysate was treated with 250 U of benzonase nuclease (Sigma) at 37 °C for 1 hr. Viral genomes were isolated using the Purelink Viral RNA/DNA mini kit and genome ratios were determined by Taqman qPCR (*Wilson et al., 2012*).

## Mouse infections and treatments

Mice were infected via the hind footpad with $1\times10^6$ PFU of A2 MuPyV. Challenge infections with A2.Δ295 were given with $1\times10^3$ PFU i.v. For comparisons of mutant viruses, mice were infected i.v. with $1\times10^6$ PFU of A2 or the mutant virus matched by g.e. For comparisons of brain infection, mice were infected i.c. with $5\times10^5$ PFU of A2 or A2.D295N/Δ297 matched by g.e. µMT mice were injected intraperitoneally (i.p.) weekly with 250 µg of 8A7H5 starting 4 dpi. For T cell depletions, mice were injected i.p. weekly with 250 µg of GK1.5 and H35-17.2 or control IgG.

## Virus infections in vitro

A31 or NMuMG cells were seeded in 12-well plates at a density of $5\times10^4$ cells/well the day before infection. Cells were washed with Iscove's Modified Dulbecco's medium with 0.1% BSA prior to infection with the specified MOI of A2 or mutant virus matched by g.e. Infections were performed at 4 °C for 1.5 hr; unbound virus was then washed out and the cells were returned to DMEM with 10% FBS. For infections with neuraminidase pretreatment, cells were incubated with or without *Vibrio cholerae* neuraminidase (Sigma) in neuraminidase buffer (PBS with 1 mM $CaCl_2$, 1 mM $MgCl_2$) at 37 °C for 30 min. Neutralization assay infections were performed at an MOI of 0.1 PFU/cell with A2 virus or VP1 mutant virus matched by g.e. For serum neutralization assays, virus was incubated with the indicated serial dilution of serum at 4 °C for 30 min prior to addition to cells. For neutralization assays with VP1 mAb, virus was incubated with 10 µg of 8A7H5 or control IgG at 4 °C for 30 min prior to addition to cells. For quantification of encapsidated genome production, A31 cells were transfected with equal amounts of WT or mutant viral DNA using Lipofectamine 3000 (ThermoFisher). At 24 hr the media was removed, and the cells were washed and placed in fresh media to remove free DNA. At 72 hr, the cells and media were collected and the amount of encapsidated genomes were quantified as above. For images of plaque formation, plaque assays were imaged 6 dpi with an Olympus IX73 inverted microscope with a QImaging Retiga 6000 Mono camera. For quantifying plaque size, plaque assays were fixed with neutral buffered formalin (NBF), stained with 1% Crystal Violet, and imaged as above. Plaque area was quantified using ImageJ (NIH).

## Viral mRNA and DNA quantification

Total RNA was isolated with TRIzol Reagent (Thermofisher) and phenol:chloroform extraction. Total cDNA was generated from 1 to 2 µg of RNA using random hexamer primers and Revertaid RT (Thermofisher). Taqman qPCR was used to quantify LT mRNA levels with normalization to TATA-Box Binding protein and compared to a standard curve to determine copy number (*Maru et al., 2017*). DNA was isolated with the Wizard Genomic DNA Purification Kit (Promega). Viral DNA was quantified by Taqman qPCR and compared to a standard curve to determine copy number (*Wilson et al., 2012*).

## TA cloning for VP1 sequencing

VP1 sequences were PCR amplified from kidney DNA and cloned using the TOPO TA cloning kit (Thermofisher). Clones were screened by restriction digest for the presence of a VP1 sequence and sequenced. VP1 sequences were screened from the presence of the D295A/N and Δ297 mutations.

## Hemagglutination assay

Viruses were diluted to $1 \times 10^7$ g.e./μL then serially two-fold diluted in PBS at the indicated pH. Virus dilutions were combined 1:1 with 0.45% sheep erythrocytes (Innovative Research) and incubated overnight at 4 °C. The highest dilution at each pH showing hemagglutination was reported as the HA titer.

## ELISA

Full-length MuPyV VP1 pentamers were kindly provided by Robert Garcea (University of Colorado, Boulder). ELISA wells were coated overnight at 4 °C with 50 ng of VP1 pentamer or $1 \times 10^7$ PFU of A2 or mutant virus matched by g.e. For avidity measurements, 8A7H5-virus complexes were treated with $NH_4SCN$ for 15 min before the addition of the secondary and detection. 8A7H5 binding was normalized for each virus to signal in the absence of $NH_4SCN$ (*Lauver et al., 2020*; *Pullen et al., 1986*).

## Flow cytometry

T cell depletions were confirmed in the peripheral blood at euthanasia by staining with antibodies for CD45, CD8α, and CD4. For quantification of kidney-infiltrating T cells, mice were injected 3 min prior to euthanasia with 3 μg of anti-CD45-FITC. T cells were isolated from the kidney by digestion with collagenase followed by centrifugation on a 44%/66% Percoll gradient. T cell numbers were determined using Flow Cytometry Absolute Count Standard beads (Bangs Laboratories). For quantification of in vitro infections, cells were trypsinized and stained with Fixable Viability Dye (ThermoFisher) followed by treatment with eBiosience Fixation/Permeabilization reagent (ThermoFisher). Cells were then stained with rat polyclonal T antigen Ab followed by an anti-rat secondary (Biolegend). For measuring sialic acid binding dependence, trypsinized cells were treated for 30 min at 37 °C in the presence or absence of *Vibrio cholerae* neuraminidase (Sigma). Bound virus was detected with rabbit polyclonal VP1 Ab followed by an anti-rabbit secondary. Samples were acquired on an LSRFortessa flow cytometer (BD Biosciences) and analyzed using FlowJo software (Tree Star).

## Immunofluorescence and histological imaging

Kidneys were immersion-fixed in NBF overnight prior to processing and paraffin-embedding. For brain preparation, mice were perfused with NBF and whole heads were fixed overnight in NBF. The brains were then removed for processing and embedding. Formalin-fixed paraffin embedded kidney and brain sections were stained with VP1, CD13 (Abcam), GFAP (Abcam), Vimentin (R&D), and THP (R&D) antibodies. Hematoxylin and eosin (H&E)-stained sagittal sections of kidneys and coronal sections of brains were evaluated by a renal pathologist and a neuropathologist, respectively, in blinded fashion. For virus binding to kidney sections, paraformaldehyde-fixed kidneys were embedded in Tissue-Tek O.C.T. Compound (Sakura) and cryosectioned. Sections were treated with *Vibrio cholerae* neuraminidase (Sigma) or buffer for 30 min at 37 °C and then incubated with virus lysate for 1.5 hr. Sections were then stained with VP1, CD13 (Abcam), and THP (R&D) antibodies. Secondary antibodies used were anti-rabbit Alexa Fluor 647 (Jackson Immunoresearch), anti-goat Alexa Fluor 488 (Jackson Immunoresearch) and Alexa Fluor 555 anti-rat secondary (Abcam). Samples were mounted with Prolong Gold Antifade Mountant with DAPI (ThermoFisher). Samples were imaged on a Leica DM4000 fluorescence microscope. For representative fluorescence images, adjustments for brightness/contrast were done uniformly to all images in the group using LAS X (Leica).

## Statistical analysis

All data are displayed as mean ± SD. The statistical tests performed are listed with the respective figures and were performed using Prism software (Graphpad). p Values of ≤ 0.05 were considered significant. Exact p values for all comparisons in figures are listed in *Supplementary file 3*. Statistical methods were not used to pre-determine sample sizes; sample sizes were determined based on the authors' experience with the model system. Sample sizes represent individual mice or biological

replicates. VP1$^+$ foci and VP1$^+$ cells were counted in a blinded fashion, blinding was not employed for other experiments. All sample sizes and number of repeats are included in the Figure Legends.

## Acknowledgements

We thank the staff of the Penn State College of Medicine Flow Cytometry Core Facility for assistance with flow cytometry experiments; the Comparative Medicine Histology Core for sample processing; Kimberly Erickson and Robert Garcea for the generous gifts of VP1 pentamers and rabbit VP1 anti-sera; and the staff of the Penn State College of Medicine Department of Comparative Medicine. This work was supported by NIH grants 5R01NS088367, 5R01NS092662, and R35NS127217. *Figure 7* was created with BioRender.com.

## Additional information

### Funding

| Funder | Grant reference number | Author |
| --- | --- | --- |
| National Institutes of Health | 5R01NS088367 | Aron E Lukacher |
| National Institutes of Health | 5R01NS092662 | Aron E Lukacher |
| National Institutes of Health | R35NS127217 | Aron E Lukacher |

The funders had no role in study design, data collection and interpretation, or the decision to submit the work for publication.

### Author contributions

Matthew D Lauver, Conceptualization, Data curation, Formal analysis, Validation, Investigation, Visualization, Methodology, Writing - original draft; Ge Jin, Katelyn N Ayers, Sarah N Carey, Charles S Specht, Catherine S Abendroth, Investigation; Aron E Lukacher, Conceptualization, Supervision, Funding acquisition, Visualization, Project administration, Writing - review and editing

### Author ORCIDs

Matthew D Lauver http://orcid.org/0000-0002-7001-9730
Katelyn N Ayers http://orcid.org/0000-0001-6156-8685
Aron E Lukacher http://orcid.org/0000-0002-7969-2841

### Ethics

This study was performed in strict accordance with the recommendations in the Guide for the Care and Use of Laboratory Animals of the National Institutes of Health. All of the animals were handled according to an approved institutional animal care and use committee (IACUC) protocol (#PRAMS201447619) of The Pennsylvania State University.

### Decision letter and Author response

Decision letter https://doi.org/10.7554/eLife.83030.sa1
Author response https://doi.org/10.7554/eLife.83030.sa2

## Additional files

### Supplementary files

• Supplementary file 1. VP1 mutations. Identity, location, and frequency of detected VP1 mutations. Superscripted numbers indicate the VP1 loop in which the mutations are located (1: BC, 2: DE, 3: EF, 4: HI). Deletions are indicated by a "Δ" followed by the deleted residues; the identity of the deleted amino acids is indicated in "()". The duplication of a residue is indicated with "dup." The presence of two mutations in a virus is indicated with "+". Sets of mutations separated by "and" indicate that both of the listed mutant viruses were isolated from the same mouse.

• Supplementary file 2. Oligonucleotide sequences. Sequences of oligonucleotides used for site-directed mutagenesis, cloning, qPCR, and sequencing.

• Supplementary file 3. Statistical information. Statistical tests used and exact p values for the comparisons of data presented in each figure as indicated.

• MDAR checklist

## Data availability

All data files are uploaded as source data files with this manuscript. Images are deposited with Dryad at https://doi.org/10.5061/dryad.prr4xgxqj.

The following dataset was generated:

| Author(s) | Year | Dataset title | Dataset URL | Database and Identifier |
|---|---|---|---|---|
| Lukacher AE, Lauver M, Jin G, Ayers K, Carey S, Specht C, Abendroth C | 2022 | T cell deficiency precipitates antibody evasion and emergence of neurovirulent polyomavirus | https://doi.org/10.5061/dryad.prr4xgxqj | Dryad Digital Repository, 10.5061/dryad.prr4xgxqj |

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
