## [Editor Report]

Progressive multifocal leukoencepalopathy (PML) is a degenerative disease of the brain that is caused by a virus in some immunocompromised patients, especially those with T-cell deficiencies. The authors use a mouse model of this virus infection to examine the components of the immune system that determine whether or not the virus will replicate and escape B-cell control. The study addresses an important question, especially given a resurgence in PML in recent years due to increasing use of immunomodulatory monoclonal antibodies to treat various diseases. The conclusions are supported by the data and confirm the expected critical role of T-cells in controlling viral early replication. The correlation of this early T-cell control with viral mutations in B-cell epitopes clarifies the relationship between proximal and distal disease causality.

---

## [Decision Letter]

**Decision letter after peer review:**

[Editors’ note: the authors submitted for reconsideration following the decision after peer review. What follows is the decision letter after the first round of review.]

Thank you for submitting the paper "T Cell Deficiency Precipitates Antibody Evasion and Emergence of Neurovirulent Polyomavirus" for consideration by *eLife*. Your article has been reviewed by three peer reviewers, one of whom is a member of our Board of Reviewing Editors, and the evaluation has been overseen by a Senior Editor.

The reviewers all agreed that the results are fascinating and important, due to the role of compromised immune systems for polyoma and indeed other viruses. Nonetheless, as you will see below, the reviewers had detailed issues with experimental justification for some of the conclusions, overly complicated prose, and organization. In the cases of immune depletion, additional controls are requested.

The policy of *eLife* is not to accept papers that require more than a few months of work. Considering the requests for additional experimentation and, just as importantly, revision of the flow and logic of the manuscript, it is not accepted for publication or revision at this point. However, the Editors may be willing to consider a revised version if comments such as requests for genetic reconstructions, confirmations of antibody depletion, and careful editing interpretations can be fully addressed.

Below, the comments of the reviewers are combined, in general, and figure-by-figure. This organization was necessary because the writing made it difficult to distinguish which experiments supported which conclusion.

Of all manuscripts submitted to eLife, only a minority will ultimately be accepted for publication. The policy of eLife is to reject manuscripts that require major revision, to facilitate the authors' submitting the manuscript elsewhere. In the case of this interesting manuscript, we encourage resubmission provided you and your colleagues are willing to address the critiques below. Otherwise, we hope that the reviews and our feedback will help you in publishing your manuscript. Thank you for giving us the opportunity to review this work.

Summary

This is an interesting study of mouse polyomavirus (MuPyV) as a model for the human pathogen JCPyV. The latter can spread to the brain in immunocompromised individuals, where replication leads to progressive multifocal leukoencephalopathy (PML). This replication is often accompanied by mutations in the major viral capsid protein, VP1, that make it resistant to certain neutralizing antibodies. The selection for such resistance can be recapitulated in the mouse model. The authors examine the contributions of the T cell response to the appearance of these resistant viruses and examine the biology of an interesting mutant that they discovered. Thus, this study addresses an important question, especially given a resurgence in PML in recent years due to the increasing use of immunomodulatory monoclonal antibodies to treat various diseases. JC polyomavirus causes Progressive Multifocal Leukoencephalopathy (PML) in T-cell immunosuppressed patients. The host-pathogen interaction behind this condition is not fully understood. Using a complicated murine infection model, the authors address the particular role of T cells in this evolution. The model includes commercially available B-cell depleted mice, first infected with MuPyV, then treated with an anti-VP1mAb, and 16 days later with IgG control or anti-CD4 and/or anti-CD8 antibodies to induce T-cell depletion. Viral replication is monitored in the kidney and blood at different times post-infection in T-cell-depleted versus control mice and the emergence of resistant variants is characterized. Finally, the fitness of a resistant variant presenting two mutations is compared in cells, tissues, and mice. The authors show that CD4 T-cells play a critical role in limiting or delaying the emergence of resistant variants. In addition, they highlight that variants that can emerge in absence of T-cell control show fitness loss in kidney but increased neurovirulence. These observations shed some light on JC virus pathogenesis in T-cell suppressed patients. The conclusions are overall supported by the data and confirm the expected critical role of T-cells in controlling viral replication. However, concerns about the presentation will require additional editing and experimental work.

Comments concerning the presentation

1. An overall concern I have is that the first part of the introduction and the entire abstract sound as though JCPyV itself is under investigation. These possibly misleading sections need to be rewritten. There is no need to be concerned that the relevance of their model is going to be questioned.

2. The relevance of these mutations relative to existing JC mutants is not made clear. In fact, two out of three reviewers missed the point that the mouse model involved a different virus from the human one. Clarified prose is needed.

3. It would be good to give some background on JCPyV receptor usage and also if the receptor(s) is (are) conserved between mouse and human and in the different tissues.

4. The last paragraph of the introduction somewhat exaggerates the findings, particularly the sentence beginning "In this study…." It is not clear that Ab-deficient mice are really the equivalent of PML patients since the patients DO produce antibodies against the virus. The model is not being questioned but it should not be oversold.

Comments concerning the figures

Figure 1: Shows that treatment with anti CD4^+^ T cell antibody, but not anti-CD8^+^ T cell antibody or IgG control antibody, reduces control of the virus in a mouse model that contains B cells.

1. The tissue should be clearly indicated in the legend and ideally on the panel.

2. The word 'resurgence' is not appropriate because the data in Figure 2B do not show early viremia.

3. The original comment from all three reviewers was that fluorescent cytometry is needed to confirm T-cell depletion, especially in the case of CD8^+^ T cells. However, it is stated somewhere that these are commercially obtained preparations. If this is the case and there is a validation of the depletions, that needs to be clearly stated. These controls are important, however they are obtained.

Supplemental Figure 1: Throughout the manuscript, the term 'blind spot' is used. This colloquial language implies a structural mechanism of neutralization that is often the case but not always.

1. Instead, more precise evolutionary language (especially for a general audience such as that of *eLife*) such as "anti-VP1 treatment allows the selection of VP1-antibody resistant viruses.

2. The data themselves show convincingly that the anti-VP1 antibody specifically neutralizes wild-type virus but not the A2Δ295 virus. At this point in the manuscript, the A2D295 virus has not been mentioned so it is not clear why it is S.Figure 1.

3. In (E), the y-axis of 'neutralizing titer' is not clear. Is it 'extent depletion'? Is it the

required serum dilution?

Figure 2 shows convincingly that the combination of anti-CD4^+^ and anti-CD8^+^ antibodies greatly reduce viral control in kidney, spleen, and brain.

1. Fluorescent cell quantitation is not provided to ensure the depletion of the relevant T cell populations,

2. The data in (B) cannot be clearly deciphered and the x-axis should be expanded.

However, the remaining panels in B, dissect the data in the time course. As in Figure 1, these data show that there is little control of the virus in the mice treated with anti-CD8^+^ antibodies.

Figure 3 is the strongest figure in the paper, showing that the sequences of viruses that grow upon anti-VP1 monoclonal antibody treatment of B-cell-deficient mice show a variety of sequence changes. Interestingly, a Δ295 and other amino acid 295 mutations are selected under all conditions and thus, as the authors conclude, confer resistance to the anti-VP1 antibody under all conditions. However, more mutations seem to be required in the absence of T cell depletion, and different spectra of mutations are found under each condition. Panel B shows the location in the VP1-antibody complex of several of the amino acids involved. Panel C shows a similar lack of neutralization for all selected viruses in cultured cells, and Panel D shows that there is different tropism and fitness for the various viruses in the kidney and spleen of infected B-cell-deficient mice in the presence of the anti-VP1 antibody.

1. It is not clear whether the mutations of interest have been installed in the otherwise wild-type virus free of other mutations that might have been selected; if it is not, this should be clearly stated.

Figure 4 shows explicit challenges with the 295 virus.

1. This virus is, presumably, reconstructed so that only this mutation is present. If it is not, it should be, and several points brought up by one of the reviewers is then relevant. Specifically, "The extensive analysis of the A2.D295N, A2.Δ297, and double mutant viruses is impressive. Given that the double mutant has enhanced replication properties, was the entire genome sequenced to confirm nothing has happened in the non-coding control region or the early region. It is not clear from the methods. Another sequencing-related question relates to the paragraph on lines 214-221: how many clones did they analyze, in order to rule out the presence of the D295N/A mutations alone? Could there have been a low-frequency present that was missed (and that might only be detected by NGS)?"

2. The legend does not describe this but the text (line 152) states that the mice used were B-cell deficient. If that is the case, this experiment with the B-cell escape variant is the same experiment done with wild-type virus previously and thus does not enhance the findings of the paper at this point, unless this reviewer is missing the point. Perhaps it was performed in the B-cell-deficient mouse supplemented with the specific antibody?

3. Why are infections done in the hind footpad in the initial protocol and then by IV for the challenge or to test the fitness of the different mutants? The infection route should be clearly described and justified in the result section.

Figure 5 switches up the viruses under study. It shows a very good experiment in which the viruses have now clearly been reconstructed in isolation. However, several inconsistencies in the data likely make it necessary to repeat several aspects of these experiments.

1. 297, D295N, and double mutant viruses were measured by somewhat indirect means such as fold increase over input and fixed time points at two different MOIs, rather than growth curves at a fixed MOI so that any differences in total yield can be clearly assessed.

2. In Panel I, the frequencies of the D295N/Δ297 double mutation in the original mice in which they were isolated, as well as a different combination of D295A/297 from another mouse, are shown. These data do not contribute mechanistically to Figure 5 and should be moved to Panel 3 where the mutations are first discussed. They do, however, provide a basis for discussion of the potential order of mutational events, which should be in the discussion.

3. The double mutant virus seems fitter than the wt. However, the abundance of this variant is very low in Fig5I, how do the authors explain that?

4. Figure 5 D to F and Figure 5 sup 1 are showing very close assays. In addition, the results are not always consistent depending on the readout: in Figure 5 D to F D295N is consistently less fit than A2 while in the supplementary figure, in panel A it looks fitter. Is it a transfected virus in the sup panel A? Why is A2 del 297 not shown in figure 5 panel A? Again, in panel E of Figure 5, this variant is as fit as wt but not in the other assay.

5. Sensitivity to neuraminidase: based on Figure 5 G and 5 sup D, del295 seems insensitive to sialidase pre-treatment, then why in panels E and F of Figure 5 sup it is sensitive?

6. With the way results are presented, it is not clear that panel H represents kidney viral loads after mice infection.

7. In figure 5H, the double mutant is as fit as the wt in the kidney. This is not reproduced in Figure 6.

Figure 6.

1. Why did the authors perform IC inoculation rather than IV or inoculation in the hind footpad as before? There are likely reasons for this, but they need to be explained.

2. The authors show in several figures that the double mutant has an advantage in cells, and this is shown here in the mouse brain. If such variants present increased neurovirulence, the authors should try to understand by what mechanism. Is binding to the receptor changed in the brain? Binding assay in brain tissues should be done in the same way as with kidney tissues.

Supplement to Figure 6 presents further exploration into the comparative phenotypes of the four viruses presented in Figure 5.

*Reviewer #1 (Recommendations for the authors):*

– In the introduction, it would be good to give some background on JCPyV receptor usage and also if the receptor(s) is(are) conserved between mouse and human and in the different tissues.

– Figures 1 and 2 should be grouped in a single and simplified figure. The experimental setup is the same and only the site of genome replication and the time of analysis differ. In figure 1, the tissue should be clearly indicated in the legend and ideally on the panel.

– Figure 4: Viral loads are not different than in absence of virus challenge in figure 3, the authors should comment.

– Why are infections done in the hind footpad in the initial protocol and then by IV for the challenge or to test the fitness of the different mutants? The infection route should be clearly described and justified in the result section.

– Figure 5 and the supplementary figure 5 are confusing and several points should be clarified in the result section:

o the double mutant virus seems fitter than the wt. However, the abundance of this variant is very low in Fig5I, how do the authors explain that?

o Figure 5 D to F and Figure 5 sup 1 are showing very close assays. In addition, the results are not always consistent depending on the readout: in Figure 5 D to F D295N is consistently less fit than A2 while in the supplementary figure, in panel A it looks fitter. Is it a transfected virus in the sup panel A? Why is A2 del 297 not shown in figure 5 panel A? Again, in panel E of Figure 5, this variant is as fit as wt but not in the other assay.

o Sensitivity to neuraminidase: based on Figure 5 G and 5 sup D, del295 seems unsensitive to sialidase pre-treatment, then why in panels E and F of Figure 5 sup it is sensitive?

o With the way results are presented, it is not clear that panel H represents kidney viral loads after mice infection.

o In figure 5H, the double mutant is as fit as the wt in the kidney. This is not reproduced in Figure 6

– Figure 6: Why did the authors perform IC inoculation rather than IV or inoculation in the hind footpad as before? Also, the authors show in several figures that the double mutant is an advantage, in cells already and here in mice brain. If such variants present increased neurovirulence, the authors should try to understand by what mechanism. Is binding to the receptor changed in the brain? Binding assay in brain tissues should be done in the same way as with kidney tissues.

*Reviewer #2 (Recommendations for the Authors):*

1. One overall concern I have is that in the first part of the introduction and the entire abstract, they make it sound as if they are studying JCPyV. These sections need to be rewritten because I think they are misleading. I don't think they need to be concerned that the relevance of their model is going to be questioned, at least not by this reviewer.

2. I found the middle of the last paragraph of the introduction to be another example of "exaggeration," particularly the sentence beginning "In this study…." I'm not sure Ab-deficient mice are really the equivalent of PML patients since the patients do produce antibodies against the virus. Again, I am not questioning their model but they should not oversell it.

3. What does "un-neutralized virus" on line 121 mean?

4. Do the authors have any sense of when during the persistent phase of the infection the mutations are arising? Is it before or after reactivation of all-out replication, for example?

5. The extensive analysis of the A2.D295N, A2.Δ297, and double mutant viruses is impressive. Given that the double mutant has enhanced replication properties, I am wondering whether they sequenced the entire genome to confirm nothing has happened in the non-coding control region or the early region. It is not clear from the methods. Another sequencing-related question relates to the paragraph on lines 214-221: how many clones did they analyze, in order to rule out the presence of the D295N/A mutations alone? Could there have been a low-frequency present that was missed (and that might only be detected by NGS)?

*Reviewer #3 (Recommendations for the authors):*

The narrative of this manuscript is difficult to follow although it contains much interesting biology. It begins by showing that T-cell control of JC polyomavirus in mice is important in the presence and absence of a B-cell response. Then, the authors show that the relevant T cells for viral control are CD4^+^, although controls for specific cell depletion are not provided. These data are presented in Figures 1, S.Figures 1, Figure 2, and one panel of Figure 3. The antibody-escape mutations in VP1 described in Figure 3 are very interesting and form the heart of this paper.

In Figure 5, four viruses are defined genetically and their properties compared in antibody-binding studies, antibody-neutralization studies, and in B-cell depleted mice. The viruses are A2 wild-type, Δ297, D295N, and double mutant. The authors show that the D295N virus (as does the Δ295 virus they have been studying up until this point) confers resistance to a particular neutralizing antibody. However, the Δ297 mutation is required to increase its fitness. Thus, Figure 5 is where all the pieces are together to ask about the properties of these viruses in the presence and absence of the neutralizing antibody and CD4^+^ T cells.

By stopping where they do in the narrative, the message of the manuscript is that compensatory mutations are often needed to increase the fitness of antibody- or drug-resistant mutations. This has been shown many times, but, as such, remains of interest to a specialized audience. ,

The writing is so confusing that suggestions, below, are to highlight the findings and make suggestions to increase their clarity and persuasiveness.

Figure 1: Shows that treatment with anti- depletion of CD4^+^ T cell antibody, but not anti-CD8^+^ T cell antibody or IgG control antibody, reduces control of the virus in mouse model that contains B cells.

1. The word 'resurgence' is not appropriate because data in Figure 2B do not show early viremia.

2. Fluorescent cytometry is needed to confirm T-cell depletion, especially in the case of CD8^+^ T cells.

Supplemental Figure 1: Throughout the manuscript, the term 'blind spot' is used. This colloquial language implies a structural mechanism of neutralization that is often the case but not always.

1. Instead, more precise evolutionary language (especially for a general audience such as that of *eLife*) such as "anti-VP1 treatment allows the selection of VP1-antibody resistant viruses.

2. The data themselves show convincingly that the anti-VP1 antibody specifically neutralizes wild-type virus but not the A2Δ295 virus. At this point in the manuscript, the A2D295 virus has not been mentioned so it is not clear why it is S.Figure 1.

3. In (E), the y-axis of 'neutralizing titer' is not clear. Is it 'extent depletion'? Is it the

required serum dilution?

Figure 2: (A) Shows a time course of viremia in B cell-deficient mice after treatment with IgG, anti-CD4^+^ , anti-CD8^+^, and a combination of anti-CD4^+^ and anti-CD8^+^ T cell antibodies. The finding that CD4^+^ , but not CD8^+^ T cells can control viremia in the absence of B cells is interesting but not pursued mechanistically.

1. Fluorescent cell quantitation is provided to ensure the depletion of the relevant T cell populations,

2. The data in (B) cannot be clearly deciphered and the x-axis should be expanded.

However, the remaining panels in B, dissect the data in the time course. As in Figure 1, these data show that there is little control of the virus in the mice treated with anti-CD8^+^ antibodies. The remaining panels show convincingly that the combination of anti-CD4^+^ and anti-CD8^+^ antibodies greatly reduce viral control in kidney, spleen, and brain.

Figure 3 is the strongest figure in the paper, showing that the sequences of viruses that grow upon anti-VP1 monoclonal antibody treatment of B-cell-deficient mice show a variety of sequence changes. Interestingly, a Δ295 and other amino acid 295 mutations are selected under all conditions and thus, as the authors conclude, confer resistance to the anti-VP1 antibody under all conditions. However, more mutations seem to be required in the absence of T cell depletion, and different spectra of mutations are found under each condition. Panel B shows the location in the VP1-antibody complex of several of the amino acids involved. Panel C shows a similar lack of neutralization for all selected viruses in cultured cells, and Panel D shows that there is different tropism and fitness for the various viruses in the kidney and spleen of infected B-cell-deficient mice in the presence of the anti-VP1 antibody.

1. It is not clear whether the mutations of interest have been installed in the otherwise wild-type virus free of other mutations that might have been selected; if it is not, this should be clearly stated.

Figure 4 shows an explicit challenge with the Δ295 virus, presumably specifically reconstructed so that only this mutation is present. Panel (B) shows viral growth following T cell depletion. The legend does not describe this but the text (line 152) that the mice used were B-cell deficient.

1. Therefore this experiment with the B-cell escape variant is the same experiment done with wild-type virus previously and thus does not enhance the findings of the paper at this point.

Figure 5 switches up the viruses under study. It shows a very good experiment in which the viruses have now clearly been reconstructed in isolation. A substitution at residue 295 (D295N), rather than the deletion that has been previously investigated, is reconstructed individually and together with a different single-amino acid deletion, Δ297, which was found in one of the natural isolates presented in Figure 3. They find (A) that only the Δ297 mutation confers neutralizing antibody resistance. This can be explained by the observed failure of either the Δ297 mutant virus or the double mutant virus to bind to the neutralizing antibody. In the other figures, the relative fitness of the wild-type A2, Δ297, D295N, and double mutant viruses were measured by somewhat indirect means such as fold increase over input and fixed time points at two different MOIs, rather than growth curves at a fixed MOI. However, the case is made convincingly that the double mutant virus grows better under all conditions in cultured cells. The authors show that these data can clearly be modified in animals, because, in the mouse kidney (H), a double mutant is present at equivalent abundance to wild-type.

1. In Panel I, the frequencies of the D295N/Δ297 double mutation in the original mice in which they were isolated, as well as a different combination of D295A/Δ297 from another mouse, are shown These data do not contribute mechanistically to Figure 5 and should be moved to Panel 3 where the mutations are first discussed. They do, however, provide a basis for discussion of the potential order of mutational events, which should be in the discussion.

Supplement to Figure 6 presents further exploration into the comparative phenotypes of the four viruses presented in Figure 5.

1. Could the authors discuss why the double-mutant virus generates the most virus but shows considerably less CPE than the D295N mutation?

Figure 6 shows pathogenesis experiments that compare wild-type and the D295N/Δ297 double mutant in B cell-deficient mice in the absence of T cell depletion. The double mutant shows similar viremia but increased growth in brain tissues. It is part of the interesting biology of this paper, that the mutations were isolated for their antibody-escape properties, but show differences in mice with no B cells. Thus, this provides the perfect jumping-off point to compare the properties of the viruses in the presence and absence of neutralizaing antiboy and T cells.

---

## [Author Response]

[Editors’ note: the authors resubmitted a revised version of the paper for consideration. What follows is the authors’ response to the first round of review.]

Comments concerning the presentation1. An overall concern I have is that the first part of the introduction and the entire abstract sound as though JCPyV itself is under investigation. These possibly misleading sections need to be rewritten. There is no need to be concerned that the relevance of their model is going to be questioned.

The abstract has been modified and the introduction has been reworked to clarify that MuPyV is used for this study.

2. The relevance of these mutations relative to existing JC mutants is not made clear. In fact, two out of three reviewers missed the point that the mouse model involved a different virus from the human one. Clarified prose is needed.

We have extensively revised the abstract and added sections to the Introduction to clarify that this study exclusively uses MuPyV, an experimental setup necessitated by the fact the JCPyV only replicates in humans. Further discussion of the nature of the mutations found in this study in comparison to those found in PML patients has been added in the Discussion (Lines 401-407), which includes the following text “The mutant viruses we identified all carried at least one mutation in the HI loop, which is also the most common site of JCPyV VP1 mutations in PML patients (Gorelik et al., 2011; Reid et al., 2011). The epitope of the VP1 mAb has significant contribution from the HI loop and competes with a large portion of the endogenous antibody response generated by MuPyV-infected mice (Lauver et al., 2020). The dominant targets of antibodies in JCPyV-infected individuals have not been determined, but given the frequency of JCPyV mutations seen in the HI loop this region may be a common target of neutralizing Ab across species.”

3. It would be good to give some background on JCPyV receptor usage and also if the receptor(s) is (are) conserved between mouse and human and in the different tissues.

A paragraph on receptor usage by JCPyV and MuPyV has been added to the Introduction (Lines 78-84). We also included a discussion of the localization of JCPyV and MuPyV binding to the distal tubules in the kidney (Lines 308-312).

4. The last paragraph of the introduction somewhat exaggerates the findings, particularly the sentence beginning "In this study…." It is not clear that Ab-deficient mice are really the equivalent of PML patients since the patients DO produce antibodies against the virus. The model is not being questioned but it should not be oversold.

In retrospect, we can appreciate how this paragraph could lead to this misperception. There is evidence (Ray et al., 2015; discussed in Lines 72-74) that PML patients have select deficiencies in their JCPyV VP1 Ab repertoire for VP1 mutations, a situation implying that a narrow VP1 repertoire sets the stage for selecting outgrowth of such variant viruses. The last paragraph of the introduction has been rewritten to clarify the experimental setup – B cell-deficient mice given an MuPyV-specific monoclonal VP1 antibody – as a model to investigate if a monospecific VP1 response will drive selection of VP1 variants (Line 88-93).

Comments concerning the figuresFigure 1: Shows that treatment with anti CD4^+^ T cell antibody, but not anti-CD8^+^ T cell antibody or IgG control antibody, reduces control of the virus in a mouse model that contains B cells.1. The tissue should be clearly indicated in the legend and ideally on the panel.

The tissue is now identified as kidney in both the legend and the figure.

2. The word 'resurgence' is not appropriate because the data in Figure 2B do not show early viremia.

We have removed this term throughout the manuscript and replaced it with “increased or elevated replication.”

3. The original comment from all three reviewers was that fluorescent cytometry is needed to confirm T-cell depletion, especially in the case of CD8^+^ T cells. However, it is stated somewhere that these are commercially obtained preparations. If this is the case and there is a validation of the depletions, that needs to be clearly stated. These controls are important, however they are obtained.

We apologize for not including this data in the original submission. Flow cytometry data confirming the depletions of CD4 and CD8 T cells in Figures 1, 2, and 4 have been added as Figure 1—figure supplement 3, Figure 2—figure supplement 1, and Figure 4—figure supplement 1.

Supplemental Figure 1: Throughout the manuscript, the term 'blind spot' is used. This colloquial language implies a structural mechanism of neutralization that is often the case but not always.1. Instead, more precise evolutionary language (especially for a general audience such as that of eLife) such as "anti-VP1 treatment allows the selection of VP1-antibody resistant viruses.

The term “blind spot” was used repeatedly to describe the inability of sera IgG from PML patients to neutralize JCPyV VP1 mutant viruses (Ray et al., 2015). We can see the reviewer’s concern with this colloquialism, and have replaced with “an inability to neutralize.”

2. The data themselves show convincingly that the anti-VP1 antibody specifically neutralizes wild-type virus but not the A2Δ295 virus. At this point in the manuscript, the A2D295 virus has not been mentioned so it is not clear why it is S.Figure 1.

Further clarification has been added to the text (Lines 110-114) regarding the use of the A2.Δ295 virus, which we previously published (Lauver et al., 2020) as a mutant virus that is resistant to neutralization by the VP1 mAb used in this study.

3. In (E), the y-axis of 'neutralizing titer' is not clear. Is it 'extent depletion'? Is it therequired serum dilution?

The y-axis and legend for supplemental-figure 1E have been updated to indicate the neutralization titer is the Log_10_ IC50 of the neutralization curves in C and D.

Figure 2 shows convincingly that the combination of anti-CD4^+^ and anti-CD8^+^ antibodies greatly reduce viral control in kidney, spleen, and brain.1. Fluorescent cell quantitation is not provided to ensure the depletion of the relevant T cell populations,

Flow cytometry data confirming T cell depletion has been added as Figure 2—figure supplement 1 (see response to comment #3 concerning the figures above).

2. The data in (B) cannot be clearly deciphered and the x-axis should be expanded.However, the remaining panels in B, dissect the data in the time course. As in Figure 1, these data show that there is little control of the virus in the mice treated with anti-CD8^+^ antibodies.

The x-axis has been expanded for Figure 2B as requested to more clearly show the time course for detection of viremia in individual mice for each of the conditions depicted in the key to the far right of this panel. This is the raw data used to generate the middle panel showing virus levels at the peak of viremia in each mouse and the right panel collating the time course of viremia per treatment group. The interpretation of this data, as the reviewer correctly points out, is that CD4 T cells rather than CD8 T cells are primarily responsible for controlling MuPyV viremia.

Figure 3 is the strongest figure in the paper, showing that the sequences of viruses that grow upon anti-VP1 monoclonal antibody treatment of B-cell-deficient mice show a variety of sequence changes. Interestingly, a Δ295 and other amino acid 295 mutations are selected under all conditions and thus, as the authors conclude, confer resistance to the anti-VP1 antibody under all conditions. However, more mutations seem to be required in the absence of T cell depletion, and different spectra of mutations are found under each condition. Panel B shows the location in the VP1-antibody complex of several of the amino acids involved. Panel C shows a similar lack of neutralization for all selected viruses in cultured cells, and Panel D shows that there is different tropism and fitness for the various viruses in the kidney and spleen of infected B-cell-deficient mice in the presence of the anti-VP1 antibody.1. It is not clear whether the mutations of interest have been installed in the otherwise wild-type virus free of other mutations that might have been selected; if it is not, this should be clearly stated.

Clarification has been added to the Results to indicate that all the mutant viruses used in this study were generated de novo by inserting the mutations identified in viremic mice into the WT MuPyV genome. In particular, the lines “To exclude possible effects of other mutations in the viral genome, we introduced several of these single and dual VP1 mutations into WT MuPyV using site-directed mutagenesis” and “We generated viruses individually carrying the D295N or Δ297 mutations in the WT A2 genome” have been added to the Results (Lines 156-158 and 189-190).

Figure 4 shows explicit challenges with the 295 virus.1. This virus is, presumably, reconstructed so that only this mutation is present. If it is not, it should be, and several points brought up by one of the reviewers is then relevant. Specifically, "The extensive analysis of the A2.D295N, A2.Δ297, and double mutant viruses is impressive. Given that the double mutant has enhanced replication properties, was the entire genome sequenced to confirm nothing has happened in the non-coding control region or the early region. It is not clear from the methods. Another sequencing-related question relates to the paragraph on lines 214-221: how many clones did they analyze, in order to rule out the presence of the D295N/A mutations alone? Could there have been a low-frequency present that was missed (and that might only be detected by NGS)?"

All VP1 mutant viruses used in the study were constructed by inserting mutations in the VP1 gene found in viremic mice into a WT genome. As noted by this reviewer, this approach was essential to exclude the possibility that changes in the readout assays could be influenced by potential mutations in other parts of the genome, such as the non-coding control region. This has been clarified in the Results as “To exclude possible effects of other mutations in the viral genome, we introduced several of these single and dual VP1 mutations into WT MuPyV using site-directed mutagenesis” (Lines 156-158).

The text and legend for Figure 5I have been updated to include that 50 VP1 clones from each mouse were sequenced (Line 254). We added the sentence “Although more sensitive sequencing methods may detect VP1 sequences with D295N/A mutations without Δ297, the high frequency of the Δ297 mutation strongly suggests this was the initial mutation” (Lines 258-260) to acknowledge the possibility that single mutations at D295 could be detected with a more sensitive sequencing technique. However, we feel that the abundance of single Δ297 mutations in the absence of clones containing an isolated D295N/A mutation strongly supports the likelihood that this was the initial mutation.

2. The legend does not describe this but the text (line 152) states that the mice used were B-cell deficient. If that is the case, this experiment with the B-cell escape variant is the same experiment done with wild-type virus previously and thus does not enhance the findings of the paper at this point, unless this reviewer is missing the point. Perhaps it was performed in the B-cell-deficient mouse supplemented with the specific antibody?

We expanded the text to state that these are B cell-deficient mice given the VP1 mAb. Further justification for the experiment has been added to the Results, in particular the text “Mice were challenged i.v. with a lower titer inoculum of the A2.Δ295 mutant virus to mimic the development of viremia with an Ab escape mutant virus. This experimental setup allowed us to separate the function of T cells in preventing the generation of VP1 mutations from the ability of T cells to control the outgrowth of a VP1 mutant virus” (Lines 175-178).

3. Why are infections done in the hind footpad in the initial protocol and then by IV for the challenge or to test the fitness of the different mutants? The infection route should be clearly described and justified in the result section.

We have added an explanation to the Results why IV infection was used for the subsequent infections comparing the mutant viruses and for the challenge experiments. In the section detailing the comparison of mutant viruses, the text “The viruses were injected i.v. to examine tropism when virus is spreading in the blood, which is the condition under which these mutations were identified, and to avoid the possibility of the viruses having impaired spread from the site of s.c. inoculation” has been added (Lines 160-163). In the section containing the challenge experiment, the sentence “Mice were challenged i.v. with a lower titer inoculum of the A2.Δ295 mutant virus to mimic the development of viremia with an Ab escape mutant virus” has been added (Lines 175-176).

Figure 5 switches up the viruses under study. It shows a very good experiment in which the viruses have now clearly been reconstructed in isolation. However, several inconsistencies in the data likely make it necessary to repeat several aspects of these experiments.1. 297, D295N, and double mutant viruses were measured by somewhat indirect means such as fold increase over input and fixed time points at two different MOIs, rather than growth curves at a fixed MOI so that any differences in total yield can be clearly assessed.

To address this concern, we expanded the time points in Figure 5F to show virus spread over time following a low MOI infection, where the double mutant virus was first detected two days before either single mutant virus and even the WT virus.

2. In Panel I, the frequencies of the D295N/Δ297 double mutation in the original mice in which they were isolated, as well as a different combination of D295A/297 from another mouse, are shown. These data do not contribute mechanistically to Figure 5 and should be moved to Panel 3 where the mutations are first discussed. They do, however, provide a basis for discussion of the potential order of mutational events, which should be in the discussion.

We appreciate the reviewer’s comment, however we feel that the results in Figure 5I are thematically distinct from the results in Figure 3. Figure 5I shows the order of the emergence of the double mutations, which logically follows once data showing differences in the functions of each of these mutations have been described in the preceding panels of Figure 5.

3. The double mutant virus seems fitter than the wt. However, the abundance of this variant is very low in Fig5I, how do the authors explain that?

We recognize that the low frequency of double mutant sequences in the samples seems to be at odds with the increased spread and replication observed in vivo. We have added further clarification to this section to address this in Lines 252-253 “Each of these mice rapidly became morbid after the detection of viremia necessitating their euthanasia within 30 days of the emergence of viremia” and Lines 251-252 “The rapid morbidity associated with the emergence of the mutation at D295 and subsequent viremia likely limited the accumulation of these double mutant viruses.”

4. Figure 5 D to F and Figure 5 sup 1 are showing very close assays. In addition, the results are not always consistent depending on the readout: in Figure 5 D to F D295N is consistently less fit than A2 while in the supplementary figure, in panel A it looks fitter. Is it a transfected virus in the sup panel A? Why is A2 del 297 not shown in figure 5 panel A? Again, in panel E of Figure 5, this variant is as fit as wt but not in the other assay.

We can now appreciate how the data in these figures would appear to be discrepant. This apparent difference is due to the timing of the assays. The assays in Figure 5D to F are short-term experiments, which demonstrate significant delays in virus production, infection, and spread by the D295N virus compared to WT virus. Figure 5-supplement 1 shows an image of a single plaque at 6 days post infection, at which point the D295N virus has been able to spread sufficiently to form plaques. The new time course data in Figure 5F shows that the D295N virus has significantly delayed ability to spread than WT virus, but does eventually spread.

5. Sensitivity to neuraminidase: based on Figure 5 G and 5 sup D, del295 seems insensitive to sialidase pre-treatment, then why in panels E and F of Figure 5 sup it is sensitive?

We now recognize that this was not fully explained in the text, and appreciate the reviewer pointing this out. Figure 5G and 5 Sup 1D show neuraminidase-insensitive binding by the D295N mutant. However, Figure 5 Sup 1E-F show that infection by the D295N mutant remains dependent on sialic acid, which indicates that the neuraminidase-insensitive binding is non-productive. Further discussion of the difference in neuraminidase sensitivity for binding vs infection by A2.D295N has been added, including “The dependence on sialic acid for infection, but not for binding, by A2.D295N indicates that by itself the D295N mutation mediates binding to a non-sialyated, non-productive receptor” (Lines 226-228).

6. With the way results are presented, it is not clear that panel H represents kidney viral loads after mice infection.

To further clarify that 5H is kidney viral loads, we added the sentence “We then assessed virus levels in the kidneys of mice infected i.v. with the mutant viruses” (Lines 239-240) to this part of the Results.

7. In figure 5H, the double mutant is as fit as the wt in the kidney. This is not reproduced in Figure 6.

Figure 5H shows similar levels of mRNA during acute (4 dpi) infection of the kidney, whereas Figure 6A-D shows pathology induced by chronic kidney infection. The similar levels of infection early but reduced infection/pathology during chronic infection by the double mutant suggest a defect in viral persistence within the kidney. The differences seen between acute and chronic kidney infection with the double mutant is now addressed in the Discussion, with “in vivo, the double mutant virus showed similar acute kidney infection to WT virus but decreased kidney infection/pathology during chronic infection (Figures 5H and 6A-D). This impairment of kidney pathology during chronic infection suggests a defect in persistence within the kidney by the D295N/Δ297 mutant virus” (Lines 385-389).

Figure 6.1. Why did the authors perform IC inoculation rather than IV or inoculation in the hind footpad as before? There are likely reasons for this, but they need to be explained.

IC inoculation is the conventional injection route for inducing MuPyV brain infection, which is not seen with s.c. or i.v. routes of inoculation. The use of IC inoculation to investigate MuPyV brain infection is now explained in the Results. “we infected WT mice intracranially (i.c.), a route of inoculation which mediates efficient MuPyV infection of the brain (Lauver et al., 2020; Mockus et al., 2020)” (Lines 272274).

2. The authors show in several figures that the double mutant has an advantage in cells, and this is shown here in the mouse brain. If such variants present increased neurovirulence, the authors should try to understand by what mechanism. Is binding to the receptor changed in the brain? Binding assay in brain tissues should be done in the same way as with kidney tissues.

Understanding the mechanism responsible for the neurovirulence of the double mutant virus is of keen interest to us as well. To this end, we found that transfection of mutant viral DNA yielded increased virus production compared to WT DNA (Figure 5—figure supplement 1B). Although this does not rule out superior/altered virus binding by the mutant, it does show that the mutant virus more efficiently completes its replication cycle at a post entry step. We speculated in the Discussion about the mechanism by which these mutations may facilitate viral assembly; however, demonstrating this would require extensive structural analysis best done in a separate study. We have invested considerable effort to carry out virus binding assays on brain sections. However, this has turned out to be a technically difficult. While MuPyV stains kidney sections well, staining of brain tissue sections is weak and nonspecific. We have attempted to improve virus binding in brain section by antigen retrieval methods, but these have proven unsuccessful. We hope the reviewer understands our inability to compare binding by the viruses in the brain due to this technical impasse.

Supplement to Figure 6 presents further exploration into the comparative phenotypes of the four viruses presented in Figure 5.

The reviewer is correct to point out that the plaque size for the double mutant is smaller. We have added a new panel to this figure supplement quantifying plaque area. Given the data indicating the double mutant produces more virus, the difference in plaque area is most likely explained by the double mutant having an increased affinity for binding host cell receptors, limiting its spread. This phenotype has previously been observed with other MuPyV mutants possessing altered receptor binding (Bauer et al. 1999). This is now noted in the Results (Lines 199-201).